# TN-SHAP-G: Graph-Structured Tensor Network Surrogates for Shapley Values and Interactions

**Farzaneh Heidari** [1 2]   **Guillaume Rabusseau** [1 2 3]

## Abstract

Shapley values are a widely used tool for attributing importance and interactions among input variables in black-box models, but their computation involves a function defined over an exponentially large space of subsets. We propose TN-SHAP-G, a framework that exploits structure in graph-structured inputs to compute Shapley values and higher-order interaction indices efficiently. Given a predictor and a fixed masking scheme, TN-SHAP-G learns a compact, graph-aligned multilinear surrogate that approximates the masked-input behavior, represented as a tensor network whose topology mirrors the input graph. Once trained from a small number of oracle queries, the surrogate enables deterministic recovery of first- and higher-order Shapley indices via the multilinear extension, without additional model queries or Monte Carlo variance. Experiments on molecular benchmarks show that the learned factorization closely matches exact Shapley values on small graphs and scales efficiently to larger graphs where sampling-based methods become infeasible.

## 1. Introduction

Explaining predictions on graph-based data requires attributing importance to individual nodes and their interactions. Shapley values (Shapley, 1953) and interaction indices (Grabisch & Roubens, 1999) offer principled, axiomatically grounded attributions widely used in ML explainability (Lundberg & Lee, 2017; Sundararajan & Najmi, 2020), yet applying them to graph predictors remains challenging: exact computation scales as $O(2^n)$ in the number of nodes, while existing approximations sacrifice essential properties. Architecture-specific methods (e.g., GNNEx-plainer (Ying et al., 2019), PGExplainer (Luo et al., 2020)) leverage message-passing but do not apply to black-box models. Sampling-based approaches (e.g., GraphSVX (Duval & Malliaros, 2021)) are model-agnostic but require thousands of forward passes, with variance worsening sharply for higher-order interactions (Fumagalli et al., 2024).

Shapley values are weighted averages over $2^n$ coalition values, forming an exponentially large table. However, this table need not be treated as unstructured: modern neural networks, including GNNs, exhibit low-rank and localized dependency patterns (Hu et al., 2022; Chen et al., 2022), and prior work has shown that induced coalition games can admit compact representations (Heidari et al., 2025). TN-SHAP-G exploits this structure by learning a factorized surrogate whose topology mirrors the input graph (Fig. 1), with nodes corresponding to tensor modes and edges to shared factors. This graph-aligned decomposition compresses the coalition-value table while making the expressivity–accuracy–cost trade-off explicit through rank parameters. Once trained from a small number of model queries, the surrogate enables deterministic recovery of Shapley values and interaction indices on the learned surrogate via closed-form integration, without imposing architectural constraints on the predictor.

**Contributions.**

- **A principled graph-aligned low-rank representation of cooperative games.** We introduce a tensor-network (Kolda & Bader, 2009) surrogate that compresses the exponential coalition-value table into a low-rank, graph-aligned multilinear representation. By matching the surrogate topology to the input graph, bond dimensions control capacity across graph separators, yielding an inductive bias aligned with local dependency patterns in graph predictors (Theorem 3.2).

- **Amortized and deterministic computation of Shapley values and interactions.** We leverage the surrogate's multilinear structure to compute Shapley values and higher-order interaction indices via closed-form Vandermonde interpolation, requiring only $O(n)$ surrogate evaluations and no Monte Carlo variance. A single trained surrogate amortizes computation of all first- and higher-order Shapley quantities. We use surrogate test $R^2$ as a principled proxy for attribution accuracy, with guarantees linking

[1]Université de Montréal, Montréal, Quebec, Canada
[2]Mila, Quebec AI Institute, Montreal, Quebec, Canada
[3]CIFAR AI Chair. Correspondence to: Farzaneh Heidari <farzaneh.heidari@mila.quebec>.

*Proceedings of the 43rd International Conference on Machine Learning*, Seoul, South Korea. PMLR 306, 2026. Copyright 2026 by the author(s).

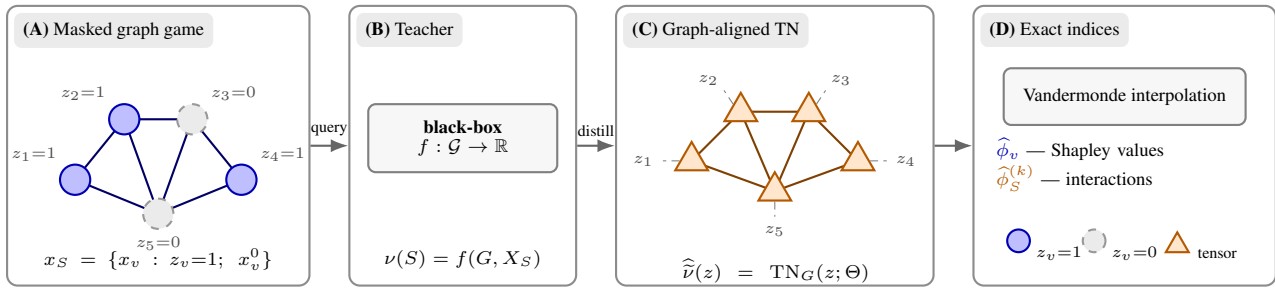

*Figure 1.* **TN-SHAP-G overview.** (A) Masked graph game with baseline replacement $(x_v^0)$. (B) Black-box teacher queried on coalitions. (C) Graph-aligned tensor network surrogate. (D) Exact Shapley values, and interactions via interpolation.

game approximation error to Shapley error (Lemma 3.7).

- **Theoretical guarantees for expressivity, correctness, and complexity.** We show that the learned surrogate uniquely determines the multilinear extension of the masked-input game, ensuring exact recovery of Shapley values and interaction indices on the surrogate. We further characterize surrogate expressivity via cut-rank arguments and prove near-linear teacher-query complexity for bounded-degree graphs (Theorems 3.2, 3.6).

- **Strong empirical accuracy and scalability.** Across molecular benchmarks, TN-SHAP-G achieves $> 0.99$ cosine similarity to exact Shapley values with as few as 50 model evaluations—10–100× fewer than SHAP-IQ and GraphSVX—while scaling to graphs where exact enumeration and sampling-based methods become impractical.

## 2. Problem Formulation

We formalize graph explanation as a cooperative game induced by node masking, which serves as the basis for all attribution methods considered in this work.

### 2.1. The Masked Graph Game

Let $G = (V, E)$ be an undirected graph with $n = |V|$ nodes, each associated with a feature vector $x_v \in \mathbb{R}^d$. Let $X = (x_v)_{v \in V} \in \mathbb{R}^{n \times d}$ denote the stacked node-feature matrix. Let $f$ be a trained graph predictor mapping attributed graphs to scalars, and let $X^0 = (x_v^0)_{v \in V} \in \mathbb{R}^{n \times d}$ be a baseline feature matrix (e.g., observational or interventional; we use the mean baseline for experiments in Section 4).

For any coalition $S \subseteq V$, define the masked feature matrix $X_S \in \mathbb{R}^{n \times d}$ by

$$(X_S)_v := \begin{cases} x_v, & v \in S, \\ x_v^0, & v \notin S, \end{cases} \tag{1}$$

and the induced cooperative game

$$\nu(S) := f(G, X_S). \tag{2}$$

Although we focus on node-level explanations (players $v \in V$), the same construction applies when players correspond

to edges or subgraphs by redefining the masking operator accordingly. Throughout, we consider explanations for a *fixed* input instance $(G, X)$, so $\nu$ is an instance-specific cooperative game over players $V$.

**Game semantics as a modeling choice.** The choice of baseline $X^0$, masking scheme (feature replacement vs. node removal), and attribution semantics (standard Shapley vs. Myerson values) are modeling decisions that define the cooperative game $\nu$. TN-SHAP-G learns a graph-aligned tensor-network parameterization of the multilinear extension of *any* such game as a function of binary coalition indicators $z \in \{0, 1\}^n$. The TN does not take raw node features as input; it models how the predictor output varies over subsets of players. Consequently, the framework is agnostic to feature type (categorical or continuous) and intervention semantics. We demonstrate this flexibility empirically in Appendices F.13–F.16.

### 2.2. Shapley Values and Interaction Indices

Given the masked graph game (2), we now define the attribution quantities of interest. The Shapley value (Shapley, 1953) measures the contribution of each node by averaging its marginal effect over all possible coalitions:

$$\phi(v) = \sum_{S \subseteq V \setminus \{v\}} \frac{|S|! \, (n - |S| - 1)!}{n!} \big[ \nu(S \cup \{v\}) - \nu(S) \big]. \tag{3}$$

Beyond first-order attributions, pairwise interaction indices quantify whether two nodes contribute synergistically or redundantly. We include both first- and second-order Shapley quantities in the experiments; formal definitions of interaction indices are deferred to Appendix D.3. Specifically, we compute the Shapley interaction index (SII) (Grabisch & Roubens, 1999). We note that $k$-SII scores (Muschalik et al., 2025) can be directly obtained from the SII values.

### 2.3. Multilinear Extension and Tensor Contraction View

Direct computation of (3) requires summing over $2^{n-1}$ coalitions. The multilinear extension (Owen, 1972) provides a continuous relaxation of discrete coalition games that transforms Shapley value computation into polynomial

integration. We show that this extension admits an exact tensor formulation, motivating the use of tensor-network surrogates when the full tensor is intractable.

Let $V = \{1, \ldots, n\}$ and let $\nu : 2^V \to \mathbb{R}$ be a set function (cooperative game). The *multilinear extension* $\widetilde{\nu} : [0,1]^n \to \mathbb{R}$ is the unique multi-affine (affine in each coordinate) polynomial in the continuous variable $z = (z_1, \ldots, z_n) \in [0,1]^n$ satisfying $\widetilde{\nu}(\mathbf{1}_S) = \nu(S)$ for all $S \subseteq V$, where $\mathbf{1}_S \in \{0,1\}^n$ denotes the indicator vector of $S$. Equivalently, for all $z \in [0,1]^n$,

$$\widetilde{\nu}(z) = \sum_{S \subseteq V} \nu(S) \prod_{v \in S} z_v \prod_{v \notin S} (1 - z_v), \qquad \widetilde{\nu}(\mathbf{1}_S) = \nu(S). \tag{4}$$

Each $z_v \in [0,1]$ continuously interpolates between excluding ($z_v = 0$) and including ($z_v = 1$) player $v$.

**Shapley values as diagonal integrals and polynomial interpolation.** Shapley values admit the classical integral representation

$$\phi(u) = \int_0^1 \frac{\partial \widetilde{\nu}}{\partial z_u}(t\mathbf{1}) \, dt, \tag{5}$$

where the equivalence to (3) follows from Owen's multilinear extension identity (Owen, 1972). Since $\widetilde{\nu}$ is multi-affine, the diagonal derivative

$$g_u(t) := \frac{\partial \widetilde{\nu}}{\partial z_u}(t\mathbf{1})$$

is a univariate polynomial of degree at most $n - 1$. Hence $\phi(u)$ is determined by finitely many evaluations of $g_u$.

Concretely, we choose probe points $t_1, \ldots, t_m \in [0,1]$ with $m \geq n$ (we use Chebyshev nodes for numerical stability), evaluate $g_u(t_j)$, and fit the unique degree-$(n-1)$ polynomial consistent with these values. Writing $g_u(t) = \sum_{k=0}^{n-1} c_{u,k} t^k$, the coefficients solve a Vandermonde-type linear system

$$\underbrace{\begin{bmatrix} 1 & t_1 & \cdots & t_1^{n-1} \\ \vdots & \vdots & & \vdots \\ 1 & t_m & \cdots & t_m^{n-1} \end{bmatrix}}_{\text{Vandermonde design}} \underbrace{\begin{bmatrix} c_{u,0} \\ \vdots \\ c_{u,n-1} \end{bmatrix}}_{c_u} = \underbrace{\begin{bmatrix} g_u(t_1) \\ \vdots \\ g_u(t_m) \end{bmatrix}}_{g_u}, \tag{6}$$

illustrated in Fig. 3. In practice we solve (6) in a numerically stable form (e.g., barycentric / least-squares when $m > n$), and then integrate the recovered polynomial in closed form:

$$\phi(u) = \int_0^1 g_u(t) \, dt = \sum_{k=0}^{n-1} \frac{c_{u,k}}{k+1}.$$

### 2.3.1. TENSOR CONTRACTION VIEW

The multilinear extension admits an equivalent tensor formulation that motivates the surrogate construction. Recall

the masked-input game $\nu : 2^V \to \mathbb{R}$ induced by a fixed graph instance $(G, X)$ with baseline $X^0$, defined as $\nu(S) = f(G, X_S)$ (Section 2.1). We encode coalitions by binary indicators $s \in \{0,1\}^n$ via $S(s) := \{v \in V : s_v = 1\}$. The following theorem gives an exact tensor contraction form of the multilinear extension.

**Theorem 2.1** (Multilinear extension as a tensor contraction). *Define the coalition tensor* $\mathcal{T} \in \mathbb{R}^{2 \times \cdots \times 2}$ *by*

$$\mathcal{T}_{s_1, \ldots, s_n} := \nu\big(S(s)\big), \qquad s \in \{0,1\}^n. \tag{7}$$

*For any* $z \in [0,1]^n$, *let* $\boldsymbol{b}_v(z_v) \in \mathbb{R}^2$ *denote the Bernoulli selector vector*

$$\boldsymbol{b}_v(z_v) := \begin{bmatrix} 1 - z_v \\ z_v \end{bmatrix}, \qquad v \in V.$$

*Then the multilinear extension* $\widetilde{\nu} : [0,1]^n \to \mathbb{R}$ *satisfies*

$$\widetilde{\nu}(z) = \sum_{s \in \{0,1\}^n} \mathcal{T}_s \prod_{v=1}^n \boldsymbol{b}_v(z_v)_{s_v}. \tag{8}$$

*Intuitively,* $\widetilde{\nu}(z)$ *is the expected value of* $\nu(S)$ *when each player* $v$ *is included independently with probability* $z_v$.

Theorem 2.1 shows that $\nu$ can be identified with an order-$n$ tensor $\mathcal{T}$ of masked evaluations of $f$ on $(G, X, X^0)$, and that the multilinear extension is exactly the contraction of $\mathcal{T}$ with Bernoulli selector vectors. The step-by-step derivation is deferred to Appendix A.3.

By uniqueness of the multilinear extension (Appendix A.1), a surrogate that fits all coalition values yields exact Shapley quantities; in practice we fit from $M \ll 2^n$ samples (Theorem 3.6) and then compute attributions deterministically from the surrogate (Section 3.3).

## 3. TN-SHAP-G

TN-SHAP-G explains black-box graph predictors by distilling the masked game $\nu$ into a graph-aligned tensor-network surrogate $\hat{\nu}$. We treat the predictor $f$ as an oracle accessible only through masked evaluations $f(G, X_S)$. The surrogate is trained to approximate the multilinear extension $\widetilde{\nu}$, from which Shapley values and interaction indices are recovered deterministically via polynomial interpolation. By uniqueness of $\widetilde{\nu}$, an exact surrogate yields exact attributions; approximation error bounds attribution error (Lemma 3.7).

Section 3.1 introduces the graph-aligned tensor-network surrogate and its expressivity, Section 3.2 describes how it is learned from $M \ll 2^n$ teacher queries, and Section 3.3 shows how Shapley values and interaction indices are recovered deterministically without further oracle access. Concretely, the method proceeds in three stages:

(i) define the masked graph game $\nu$;

(ii) learn a graph-aligned TN surrogate $\hat{\nu}$ approximating the multilinear extension $\widetilde{\nu}$;

(iii) extract Shapley values and interaction indices from $\hat{\nu}$.

**Algorithm 1 TN-SHAP-G** (single-surrogate deterministic O1 Shapley)

---

**Require:** fixed instance $(G = (V, E), X)$, baseline $X^0$, black-box teacher $f$, query budget $M$, TN bond dim $\chi$, probe points $\{t_j\}_{j=1}^m$

**Ensure:** node Shapley values $\{\hat{\phi}(u)\}_{u \in V}$

1: Sample coalitions $\mathcal{S} = \{S_j\}_{j=1}^M \sim \mathcal{D}$
2: Query labels $y_j \leftarrow f(G, X_{S_j})$ for all $j$
3: Train graph-aligned TN surrogate $\hat{\nu}(\cdot; \Theta)$ by minimizing Eq. (13)
4: **for** $j = 1, \ldots, m$ **do**
5:      Set all sites to $\boldsymbol{b}_v(t_j)$ and contract TN on $G$ to obtain reusable messages
6:      **for** each $u \in V$ **do**
7:          Compute $g_u(t_j) = \partial_{z_u} \hat{\nu}(t_j \mathbf{1})$ by replacing $\boldsymbol{b}_u(t_j) \mapsto \boldsymbol{b}'_u(t_j)$
8:      **end for**
9: **end for**
10: For each $u$, interpolate $g_u(t)$ from $\{(t_j, g_u(t_j))\}_{j=1}^m$ and return $\hat{\phi}(u) = \int_0^1 g_u(t)\, dt$

---

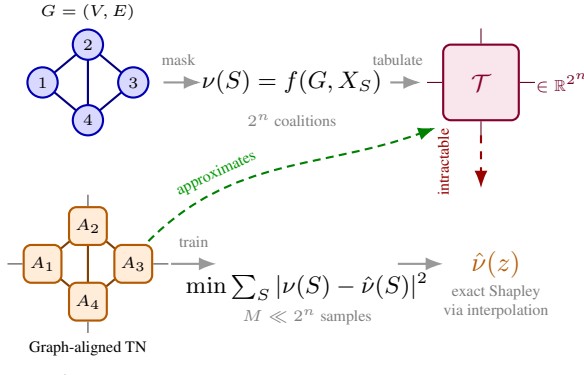

*Figure 2.* From graph game to graph-aligned tractable surrogate.

### 3.1. Graph-Aligned Surrogates and Rank-Based Expressivity

The coalition tensor $\mathcal{T} \in \mathbb{R}^{2 \times \cdots \times 2}$ defined in (8) contains $2^n$ entries, and constructing it explicitly would require $2^n$ black-box evaluations of $f(G, X_S)$, which is prohibitive even for moderate $n$. We therefore construct a compact surrogate by imposing a tensor network (TN) structure on $\mathcal{T}$ whose topology mirrors the input graph $G = (V, E)$. We treat each node $v \in V$ as a player. For each node $u \in V$, we introduce a core tensor

$$\mathcal{A}^{(u)} \in \mathbb{R}^{2 \times \chi \times \cdots \times \chi}, \qquad (9)$$

of order $|\mathcal{N}(u)| + 1$ with one *physical* index of dimension 2 encoding coalition membership $s_u \in \{0, 1\}$ and $|\mathcal{N}(u)|$ *bond* indices of dimension $\chi$, one per neighbor $v \in \mathcal{N}(u)$. Contracting bond indices along edges $(u, v) \in E$ and fixing

physical indices to $s \in \{0, 1\}^n$ defines a surrogate coalition tensor:

$$\widehat{\mathcal{T}}_s = \sum_{\{\alpha_e\}_{e \in E}} \prod_{u \in V} \mathcal{A}^{(u)}_{s_u, \alpha_{\mathcal{N}(u)}}, \qquad (10)$$

where $\alpha_{\mathcal{N}(u)} = (\alpha_{(u,v)})_{v \in \mathcal{N}(u)}$ collects the bond indices incident to node $u$, each summed over $[\chi] = \{1, \ldots, \chi\}$. The contraction sums over all bond indices and returns a scalar for each coalition assignment $s \in \{0, 1\}^n$, defining an implicit approximation $\widehat{\mathcal{T}}$ of the coalition tensor. Intuitively, bond indices transmit interaction information between neighboring nodes, while the bond dimension $\chi$ controls the capacity of this information flow across graph separators. This expressivity can be formalized via the *matricization rank* of the represented tensor across a bipartition. Fig. 2 shows the compression from $2^n$ coalition values to a $\chi$-controlled surrogate.

The graph-aligned architecture encodes a locality assumption: interaction effects between distant nodes must propagate through intermediate bonds in $G$, with information capacity bounded by $\chi$ per edge. This bias is well-suited whenever the masked-input game exhibits predominantly local, low-rank dependencies over the graph, an assumption satisfied by many practical predictors (including but not limited to message-passing GNNs). When the underlying game exhibits dense long-range interactions that do not respect graph distance, larger bond dimensions or alternative TN topologies may be required.

The expressivity of a tensor network is governed by its ability to capture correlations across partitions. This is formalized by the matricization rank, which measures the complexity of dependencies between two groups of variables.

**Definition 3.1** (Matricization rank). For a tensor $\mathcal{T}$ over vertex set $V$ and a bipartition $(A, B)$ of $V$, let $\mathrm{rank}_{A|B}(\mathcal{T})$ denote the rank of the matrix obtained by grouping indices in $A$ as rows and indices in $B$ as columns.

This quantity, also known as Schmidt rank in quantum information, measures correlation complexity across a partition (Levine et al., 2019). The following theorem makes explicit that any TN must allocate sufficient bond dimension to capture these correlations.

**Theorem 3.2** (Cut-rank lower bounds separator capacity). *Let $\mathcal{T} \in \mathbb{R}^{2 \times \cdots \times 2}$ be the coalition tensor on vertex set $V$, and consider a TN representation of $\mathcal{T}$. Suppose that removing a set of TN edges $\delta(A, B)$ separates the cores indexed by $A$ from those indexed by $B$. If each edge $e \in \delta(A, B)$ has bond dimension $\chi_e$, then*

$$\mathrm{rank}_{A|B}(\mathcal{T}) \leq \prod_{e \in \delta(A,B)} \chi_e. \qquad (11)$$

*In particular, if a single bond of dimension $\chi$ separates $A$ and $B$, then $\chi \geq \mathrm{rank}_{A|B}(\mathcal{T})$.*

*Proof.* See Appendix C.1. □

As a consequence, compact graph-aligned surrogates exists only when the coalition tensor has low matricization rank across graph-induced separators.

**Corollary 3.3** (Necessary condition for compact graph-aligned surrogates). *If a graph-aligned TN on $G$ uses uniform bond dimension $\chi$ on every edge, then for every separator $(A, B)$ induced by cutting a set of TN edges $\delta(A, B)$,*

$$\text{rank}_{A|B}(\mathcal{T}) \leq \chi^{|\delta(A,B)|}. \tag{12}$$

This bound implies that graph-aligned TNs can achieve a target fidelity with smaller bond dimension, hence fewer parameters and queries, than chain-based surrogates such as tensor trains (TT) (Oseledets & Tyrtyshnikov, 2010) when $\mathcal{T}$ has low cut-rank across graph separators. See the proof in Appendix C.1.

**Relationship to general TN theory.** The principle that TN expressivity depends on cut-rank across separators is well established in the TN literature. Our contribution is to instantiate this principle for the *coalition tensor of graph-induced cooperative games*, showing that (i) graph separators in $G$ control the cut-rank of the masked-input game (Theorem 3.2, Corollary 3.3, Theorem C.1), (ii) graph-aligned TNs achieve strictly better expressivity–parameter tradeoffs than chain factorizations for such games, and (iii) this translates to practical improvements in Shapley recovery under matched parameter budgets (Section 4, Fig. 6). TN-SHAP-G builds on TN-SHAP (Heidari et al., 2025), which introduced TN surrogates for Shapley computation on general (non-graph) inputs using tensor-tree topologies; the present work introduces graph-aligned TN topologies and demonstrates their theoretical and empirical advantages for graph-structured games.

**Example: cycle graphs and TT rank inflation.** This separation–capacity effect is easiest to see on a cycle (ring) graph. If $G$ is a ring, the graph-aligned surrogate corresponds to a tensor-ring (TR) whose separators align with the graph, so bond dimension $\chi$ suffices.

A tensor-train (TT/MPS), however, enforces a 1D chain and only exposes prefix–suffix cuts. Even when $\mathcal{T}$ has an efficient TR representation with bond dimension $\chi$, the induced TT ranks can generically grow to $\Omega(\chi^2)$, increasing parameter count (and thus query/sample needs) to reach the same fidelity. The following theorem quantifies this inflation.

**Theorem 3.4** (TT-rank inflation from tensor-ring structure). *Let $\mathcal{T}$ admit a tensor-ring decomposition with uniform rank $\chi$. Then any tensor-train representation of $\mathcal{T}$ generically requires TT-ranks $\Omega(\chi^2)$, and approximation with smaller ranks incurs error lower-bounded by the truncated singular value tail of the corresponding matricization.*

See the proof in Appendix C.2.

This rank inflation is not an artifact of cycle graphs: for GNN predictors, similar effects arise from the interaction structure itself. For message-passing GNN predictors, recent theory shows that interaction complexity across a vertex partition $(A, B)$ is governed by a graph-theoretic *walk index* of the cut boundary (Razin et al., 2023). In current notation, this implies that $\text{rank}_{A|B}(\mathcal{T})$ can be large for cuts whose boundary admits many length-$(L-1)$ walks crossing the partition, even for moderate depth $L$. Combined with Theorem 3.2, this predicts that chain surrogates (TT/MPS) may require large ranks for unfavorable orderings, while graph-aligned TNs explicitly allocate capacity along graph separators and are ordering-invariant by construction.

**Corollary 3.5** (TT ordering sensitivity for GNN-induced games). *For a TT/MPS surrogate with node ordering $\pi$, every chain cut $(A, B)$ along $\pi$ satisfies $\rho_k \geq \text{rank}_{A|B}(\mathcal{T})$ by Theorem 3.2. Thus, on graphs where the GNN induces high cut interaction rank for some partitions (e.g., high-walk-index cuts (Razin et al., 2023)), TT can require large ranks for certain orderings, while graph-aligned TNs remain ordering-invariant by construction. See the full proof and the formal definition in Appendix C.2.*

Overall, this section suggests that graph-aligned surrogates can reach a target fidelity with smaller bond dimension (hence fewer parameters and teacher queries) when interactions are localized and cut-ranks across graph separators are low. Section 4 supports this empirically via parameter-matched comparisons of predictive accuracy and O1/O2 Shapley recovery. Next, we present the training procedure and efficient Shapley/interaction recovery from the learned surrogate.

### 3.2. Tensor Network Surrogate Training

Having established the expressivity advantages of graph-aligned surrogates, we now describe how to learn them from limited oracle access.

We train $\hat{\nu}$, parameterized by a graph-aligned TN, to approximate the masked graph game $\nu(S)$ using teacher queries on sampled coalitions. The training dataset $\mathcal{S} = \{S_j\}_{j=1}^M$ consists of coalitions $S \subseteq V$ labeled with teacher evaluations $y_j = \nu(S_j)$. Since the full coalition space has size $2^n$, we sample from a distribution that balances coverage across coalition sizes while prioritizing coalitions most informative for Shapley and low-order interactions (details in Appendix D). In practice, uniform coalition sampling already yields competitive surrogate quality; however, structure-aware sampling that enriches the training set with edge-flip and triangle-flip neighbors of base coalitions substantially improves higher-order interaction recovery (O3 cosine from $0.48$ to $0.91$; see Appendix F.17).

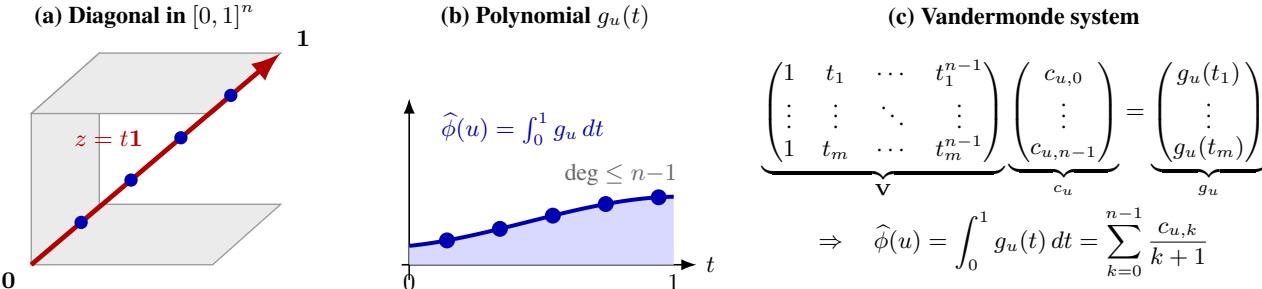

**(a) Diagonal in $[0,1]^n$**           **(b) Polynomial $g_u(t)$**           **(c) Vandermonde system**

**Figure 3.** **Deterministic Shapley from a TN surrogate via diagonal interpolation.** (a) Restrict to the diagonal $z = t\mathbf{1}$ and evaluate $g_u(t) = \partial_{z_u}\hat{\nu}(t\mathbf{1})$ at probe points $t_j$. (b) Multilinearity implies $g_u$ is a degree-$\leq n{-}1$ polynomial. (c) Interpolate coefficients via a Vandermonde design (or a stable equivalent) and integrate exactly: $\widehat{\phi}(u) = \int_0^1 g_u(t)\,dt$.

### 3.2.1. DISTILLATION

We train a graph-aligned TN surrogate $\hat{\nu}(\cdot;\Theta)$ : $[0,1]^n \to \mathbb{R}$ to approximate the masked game $\nu$ using $M$ teacher queries. Coalitions $\mathcal{S} = \{S_1, \ldots, S_M\}$ are sampled from a distribution $\mathcal{D}$ that balances coverage across coalition sizes while prioritizing coalitions informative for low-order interactions (details in Appendix D). The surrogate is fit by empirical risk minimization:

$$\min_{\Theta} \; \frac{1}{M}\sum_{j=1}^{M}\big(\nu(S_j) - \hat{\nu}(\mathbf{1}_{S_j};\Theta)\big)^2, \tag{13}$$

where $\hat{\nu}(\mathbf{1}_S;\Theta)$ denotes the TN output at the binary indicator $\mathbf{1}_S$, selecting the included/excluded slice at each node according to $S$.

The learned surrogate is the multilinear function induced by the TN parameters $\Theta$:

$$\hat{\nu}(z;\Theta) := \widehat{\mathcal{T}}(\Theta) \times_1 \boldsymbol{b}_1(z_1) \times_2 \cdots \times_n \boldsymbol{b}_n(z_n), \tag{14}$$

where $\widehat{\mathcal{T}}(\Theta)$ is the surrogate coalition tensor implicitly represented by the TN cores, $\boldsymbol{b}_v(z_v) = [1 - z_v, \; z_v]^\top$ are Bernoulli selector vectors, and $\times_i$ denotes mode-$i$ contraction (Appendix A). The contraction is performed directly on the factorized representation without materializing the $2^n$-entry tensor. At binary inputs $z = \mathbf{1}_S$, evaluating $\hat{\nu}(\mathbf{1}_S)$ reduces to contracting the TN with physical indices fixed by coalition membership.

The following theorem, adapted from Khavari & Rabusseau (2021, §4), shows that the required query budget scales with TN parameter count rather than $2^{|V|}$

**Theorem 3.6** (Near-linear teacher-query complexity on bounded-degree graphs). *Let $\mathcal{H}_G$ be a tensor-network hypothesis class on topology $G = (V, E)$ with parameter count $N_G = \sum_{v \in V}\prod_{e \in E_v}\dim(e)$ (Khavari & Rabusseau, 2021, §4). Assume $\nu(S) \in [-B, B]$ and let $\hat{\nu} \in \mathcal{H}_G$ be learned by empirical risk minimization (ERM) from $M$ i.i.d. teacher queries $(S_j, \nu(S_j))$. Then with probability $\geq 1 - \delta$,*

$$\mathcal{E}(\hat{\nu}) - \inf_{h \in \mathcal{H}_G} \mathcal{E}(h) \; = \; \tilde{O}\!\left(\frac{B^2\,(N_G + \log(1/\delta))}{M}\right),$$

*where $\mathcal{E}(h) = \mathbb{E}\big[(h(S) - \nu(S))^2\big]$ and $\tilde{O}(\cdot)$ hides logarithmic factors in $M$ and $|V|$. Moreover, if $\deg_{\max}(G) \leq \Delta$ and $\dim(e) \leq \chi$, then $N_G \leq |V|\chi^\Delta$, hence $M = \tilde{O}(|V|\chi^\Delta/\varepsilon^2)$ suffices to reach excess risk $\leq \varepsilon^2$, i.e., near-linear in $|V|$ for fixed $\chi, \Delta$.*

See Appendix C.4 for the details and the full proof.

Theorem 3.6 shows that graph-aligned TN surrogates can be learned from a near-linear number of teacher queries; we now quantify how this sample efficiency translates into surrogate fidelity on unseen coalitions and, in turn, into Shapley attribution accuracy.

**Test $R^2$ as a fidelity metric.** We measure surrogate fidelity by held-out $R^2$ on coalitions in $\mathcal{S}_{\text{test}}$:

$$R^2 \; = \; 1 - \frac{\sum_{S \in \mathcal{S}_{\text{test}}}\big(\nu(S) - \hat{\nu}(\mathbf{1}_S)\big)^2}{\sum_{S \in \mathcal{S}_{\text{test}}}\big(\nu(S) - \bar{\nu}\big)^2}.$$

Theorem 3.6 links the teacher-query budget $M$ and TN capacity to surrogate generalization, while Lemma C.2 (Appendix C.7) highlights an intrinsic approximation floor when the chosen TN topology or bond dimension $\chi$ cannot capture the game's cut-rank structure (quantified by truncated-SVD residuals of matricizations). Thus, increasing $M$ and/or $\chi$ typically improves $R^2$ until this floor is reached. Finally, Lemma 3.7 guarantees that uniform game approximation $\|\nu - \hat{\nu}\|_\infty \leq \varepsilon$ implies worst-case Shapley error at most $2\varepsilon$. Empirically, we observe a strong correlation between test $R^2$ and Shapley cosine similarity (Fig. 5), making $R^2$ a practical proxy and tuning knob for attribution accuracy.

**Lemma 3.7** (Stability of Shapley values and interactions under game perturbations). *Let $\nu, \hat{\nu} : 2^V \to \mathbb{R}$ satisfy $\|\nu - \hat{\nu}\|_\infty \leq \varepsilon$. Then for all $i \in V$, $|\phi_i(\nu) - \phi_i(\hat{\nu})| \leq 2\varepsilon$. Moreover, for any $T \subseteq V$ with $|T| = k \geq 1$, the Shapley interaction index satisfies $|\phi_T(\nu) - \phi_T(\hat{\nu})| \leq 2^k\varepsilon$.*

See Appendix C.3 for the full proof.

Together, these results justify computing Shapley values and interaction indices directly from the learned TN surrogate: once the surrogate achieves high fidelity, the induced

attributions are guaranteed to be accurate. We now show how Shapley quantities can be computed deterministically from the multilinear TN representation.

### 3.3. Deterministic Shapley and Interactions from the TN Surrogate

Recall $\hat{\nu}$ is the trained graph-aligned TN surrogate. Since $\hat{\nu}$ is a multilinear polynomial in $z$ by construction, we can compute Shapley values and interaction indices *deterministically* from the surrogate, without further teacher evaluations.

For any player $u \in V$, Shapley values admit the diagonal-derivative integral representation (Owen, 1972)

$$\widehat{\phi}(u) = \int_0^1 \frac{\partial \hat{\nu}}{\partial z_u}(t\mathbf{1})\, dt. \tag{15}$$

The integrand is evaluated by TN contraction. At a given $t \in [0, 1]$, we contract the surrogate at the diagonal input $z = t\mathbf{1}$, using $\boldsymbol{b}_v(t) = [1 - t,\ t]^\top$ at every node $v$, except that at the queried node $u$ we insert the derivative vector $\boldsymbol{b}'_u(t) = [-1,\ 1]^\top$. Higher-order interactions are obtained by inserting $\boldsymbol{b}'_{u_i}(t)$ at multiple sites (Appendix B).

Define $g_u(t) := \partial_{z_u} \hat{\nu}(t\mathbf{1})$. Multilinearity implies $\deg(g_u) \leq n-1$, so $\widehat{\phi}(u) = \int_0^1 g_u(t)\, dt$ is recovered from $m = O(n)$ probe points via the diagonal interpolation recipe in Section 2.3 (Fig. 3). Each evaluation $g_u(t_j)$ is obtained by the same local-replacement contraction described above, i.e., $\boldsymbol{b}_u(t_j) \mapsto \boldsymbol{b}'_u(t_j)$ while $\boldsymbol{b}_v(t_j)$ is used for $v \neq u$. Mixed partial derivatives for interactions are handled identically by replacing multiple sites (Appendix B).

Notably, the same surrogate supports *all* first- and higher-order quantities, whereas sampling-based estimators typically require separate Monte Carlo budgets for each order.

#### 3.3.1. Computational Complexity

Given a trained surrogate, the cost of computing Shapley values and interaction indices is dominated by tensor network contraction and polynomial interpolation. The graph-aligned structure of the surrogate allows both first- and second-order attributions to be computed efficiently by exploiting locality in the interaction graph.

**Theorem 3.8** (Amortized runtime on bounded-treewidth graphs). *Let $\hat{\nu}$ be a graph-aligned TN surrogate on $G = (V, E)$ with bond dimension $\chi$. Assume we contract the TN using a tree decomposition of width $\tau$ (max bag size $\tau + 1$), and reuse the resulting messages to form single-node and single-edge environments. Then for a fixed probe point $t \in [0, 1]$:*

  *(i) all first-order derivatives $\{\partial_{z_u} \hat{\nu}(t\mathbf{1})\}_{u \in V}$ can be computed in $O(|V|\chi^{\tau+1})$ total time;*

  *(ii) all edge mixed derivatives $\{\partial_{z_u} \partial_{z_v} \hat{\nu}(t\mathbf{1})\}_{(u,v) \in E}$ can be computed in $O(|V|\chi^{\tau+1})$ total time.*

*Therefore, using $m$ probe points, computing all Shapley values and all edge interaction indices costs $O(m|V|\chi^{\tau+1})$ time.*

*Table 1.* Correctness on small graphs (mean $\pm$ std). GraphSVX exhibits high variance due to poor convergence; O2 is omitted as GraphSVX does not support interaction indices.

| Method | Dataset | O1 | O2 |
|---|---|---|---|
| TN-SHAP-G | Benzene | 0.9979 (.004) | 0.9612 (.051) |
| GraphSVX | Benzene | 0.30 (.51) | — |
| TN-SHAP-G | Mutagenicity | 0.9933 (.012) | 0.9688 (.071) |
| GraphSVX | Mutagenicity | 0.45 (.66) | — |

See proof in Appendix C.6.

*Remark* 3.9 (Message reuse across probe points). The tree decomposition structure is independent of the probe point $t$. However, the message values depend on $t$ through the Bernoulli vectors $\boldsymbol{b}_v(t)$, so messages cannot be directly reused across different probe points. The stated complexity $O(m|V|\chi^{\tau+1})$ thus reflects $m$ independent message-passing sweeps. In practice, the overhead is modest since $m = O(n)$ and the per-sweep cost is linear in $|V|$ for bounded treewidth.

**Corollary 3.10** (Constant treewidth and bond dimension). *If $\tau = O(1)$ and $\chi = O(1)$, then the total cost is linear in graph size: $O(m|V|)$ time for all Shapley values and edge interactions.*

Many molecular graphs encountered in practice admit low-width decompositions, making the above bound effective.

## 4. Experiments

We evaluate **TN-SHAP-G** along three axes: (i) correctness on small graphs where exact enumeration is feasible, (ii) query efficiency relative to sampling-based estimators, and (iii) scaling with graph size. Unless stated otherwise, we explain a fixed test instance $(G, X)$ using the node-masking game in (2) with a mean-feature baseline. We report surrogate fidelity via held-out coalition $R^2$ and attribution fidelity via cosine similarity to enumeration on small graphs ($n \leq 20$) for O1 Shapley values and O2 interactions. Query efficiency is evaluated against permutation sampling (Castro et al., 2009) and SHAP-IQ (Fumagalli et al., 2024) as representative model-agnostic Shapley estimators (GraphSHAP-IQ (Muschalik et al., 2025) in F.12), and evaluate on three molecular/ protein benchmarks including Mutagenicity (Debnath et al., 1991), Benzene (Sanchez-Lengeling et al., 2020), and PROTEINS (Dobson & Doig, 2003); full experimental details in Appendix E.

**Correctness on Small Graphs & query efficiency.** On small graphs ($n \leq 20$), TN-SHAP-G closely matches exact Shapley values (O1) and interaction indices (O2) across molecular benchmarks (Table 1). Under matched teacher-query budgets, TN-SHAP-G achieves high attribution accuracy with 10–100$\times$ fewer model queries than permutation sampling and SHAP-IQ (Fig. 4). Notably,

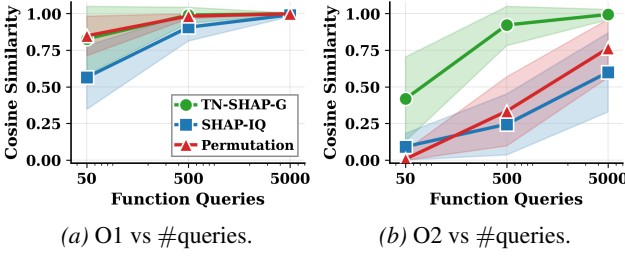

*(a)* O1 vs #queries.    *(b)* O2 vs #queries.

*Figure 4.* **Query efficiency.** TN-SHAP-G reaches 0.95 cosine similarity with 10–100× fewer queries than baselines; a single surrogate yields both O1 and O2.

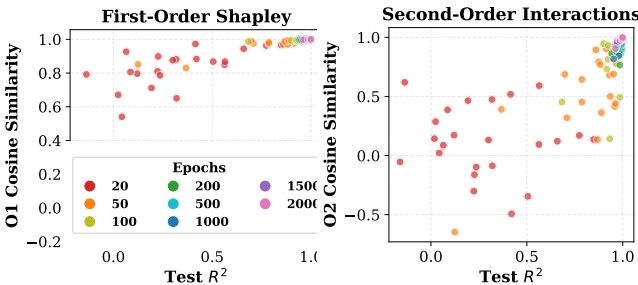

*(a)* O1 cosine vs. surrogate $R^2$.    *(b)* O2 cosine vs. surrogate $R^2$.

*Figure 5.* **Surrogate $R^2$ predicts Shapley accuracy.** Each point is one (graph, epoch) pair; color indicates training progress.

a single learned surrogate supports both O1 and O2 recovery, whereas sampling-based methods require separate Monte Carlo budgets and exhibit high variance, particularly for higher-order interactions. Code is available at `https://github.com/farzana0/TN-SHAP-G`.

**Fidelity, scaling, and structure ablation.** Surrogate test $R^2$ strongly correlates with O1 and O2 cosine similarity across training (Fig. 5), validating $R^2$ as a practical proxy for Shapley accuracy and supporting the theoretical stability guarantees. TN-SHAP-G scales efficiently to larger graphs (the coarsening architecture; Appendix F.7), with practical training and deterministic attribution times (Fig. 7), enabling explanations beyond the reach of enumeration or sampling. Finally, structure ablations show that graph-aligned TN surrogates consistently outperform tensor trains under parameter matching, with TT additionally sensitive to node ordering (Fig. 6). Details of the graph-aligned scaling architecture used in these experiments are deferred to Appendix F.4. TN-SHAP-G pays a one-time training cost per instance, after which all Shapley quantities are extracted from the same surrogate without further model queries. Table 2 compares amortized total time against GraphSHAP-IQ on 20 matched Mutagenicity graphs ($n \leq 20$). In the low-budget GPU regime (mean training time 4.12 s), TN-SHAP-G becomes faster at $K \geq 5$ repeated queries, with O1 extraction costing 0.010 s and O2 extraction 0.119 s per query, compared to 2.08 s per GraphSHAP-IQ query (Appendix F.18).

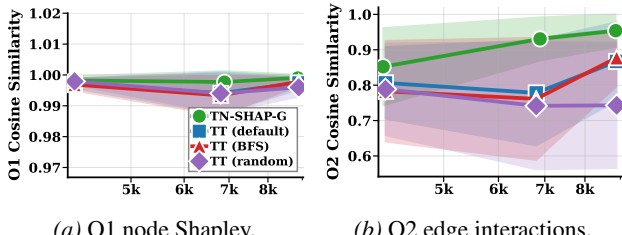

*(a)* O1 node Shapley.    *(b)* O2 edge interactions.

*Figure 6.* **Structure ablation: graph-aligned TN vs. tensor train (TT).** Attribution accuracy under parameter matching and a low query budget (10 n coalitions per graph). Different colors correspond to Different TT node orderings.

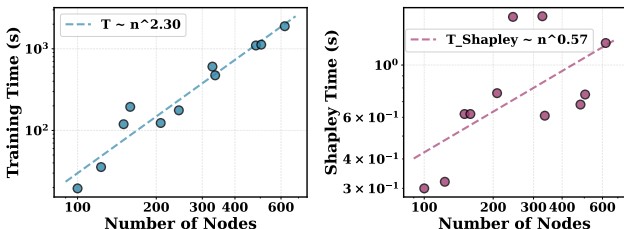

*(a)* Training time for learning the graph-aligned TN surrogate as a function of graph size.    *(b)* Time to compute all node Shapley values from the trained surrogate via TN-SHAP-G.

*Figure 7.* **Scaling behavior of TN-SHAP-G.**

*Table 2.* Amortized O1/O2 timing: TN-SHAP-G vs. GraphSHAP-IQ on 20 matched Mutagenicity graphs ($n \leq 20$). $K$ = number of repeated queries on the same graph instance.

| $K$ | TN-SHAP-G (s) | GraphSHAP-IQ (s) | TN wins |
|-----|---------------|------------------|---------|
| 1 | 197.8 | 11.6 | 0/20 |
| 5 | 206.2 | 57.9 | 1/20 |
| 20 | 237.8 | 231.6 | 6/20 |
| 50 | 301.1 | 579.1 | 17/20 |
| 100 | 406.7 | 1158.1 | 18/20 |

## 5. Related Work

**Tensor networks and query-efficient learning.** Tensor networks compress high-dimensional functions with complexity governed by bond dimension rather than input dimension (Kolda & Bader, 2009; Oseledets & Tyrtyshnikov, 2010). Tensor cross interpolation (TCI) (Oseledets & Tyrtyshnikov, 2010; Fernández et al., 2025) enables learning TN representations from few queries, and topology-aligned networks can outperform tensor trains on structured functions (Tindall et al., 2024). TN-SHAP-G builds on these ideas to learn *graph-aligned multilinear* surrogates and compute Shapley values exactly on the surrogate via the Owen extension (Owen, 1972). In closely related work, Marzouk et al. (2026) show that tensor trains enable efficient computation of Shapley values.

**Shapley values on graphs.** Shapley explanations (Shapley, 1953) are widely used in tabular ML (e.g., KERNELSHAP (Lundberg & Lee, 2017)) and extended to interactions via indices such as Shapley–Taylor (Sundararajan & Najmi, 2020). On graphs, GRAPHSVX (Duval & Malliaros, 2021) adapts kernel-based Shapley estimation by sampling masked graphs. However, sampling-based estimators often require many model evaluations and incur variance that grows with interaction order.

**Surrogate-based attribution.** Surrogate modeling amortizes explanations by approximating a black-box function under perturbations (e.g., LIME (Ribeiro et al., 2016), KernelSHAP (Lundberg & Lee, 2017)). Unlike generic linear surrogates, TN-SHAP-G uses structure-aware multilinear TN surrogates, enabling deterministic Shapley and interaction computation without Monte Carlo estimation.

**Proxy-based Shapley estimation.** The Regression MSR principle (Witter et al., 2026) and ProxySPEX (Butler et al., 2026) fit a proxy model to the value function and use a residual-correction scheme to obtain unbiased Shapley estimates. TN-SHAP-G takes a complementary approach: rather than fitting a generic proxy, it learns a graph-aligned TN representation of the multilinear extension itself. In principle, the learned TN could serve as the proxy in an MSR-style pipeline; we leave this for future work. TreeSHAP (Bifet et al., 2022) and TreeSHAP-IQ (Muschalik et al., 2024) derive exact Shapley computation from the polynomial structure of tree models via dynamic programming. While both viewpoints involve polynomial structure, the connection is high-level: TN-SHAP-G exploits a low-rank factorization of the multilinear extension, whereas TreeSHAP exploits the combinatorial structure of decision-tree paths.

## 6. Limitations and Future Work

The current implementation assumes a fixed baseline masking scheme and focuses on node-based players, though the framework extends to edge- and motif-level players. Higher-order interaction accuracy can be sensitive to surrogate misspecification even when test $R^2$ is high; improving O2 and above may benefit from tailored coalition sampling, explicit regularization of mixed partials, or adaptive rank allocation. Future work includes adaptive decomposition choices (tree decompositions or learned TN topologies), richer coalition distributions (connected subgraphs or domain-specific masks), and extensions to other structured modalities beyond graphs.

## 7. Conclusion

We presented TN-SHAP-G, a framework for computing Shapley values and interactions by learning a graph-aligned tensor network surrogate. By exploiting low-rank structure in value-functions, TN-SHAP-G enables deterministic attri-

bution recovery with $O(n)$ queries on the learned surrogate. Future work includes adaptive rank and extension to other structures.

## Acknowledgements

Farzaneh Heidari thanks Farzan Heidari for constructive discussions and careful proofreading of the paper.

## Impact Statement

This work advances methods for explaining black-box graph predictors through principled game-theoretic attributions. The primary societal benefit lies in improving transparency of graph neural networks deployed in high-stakes domains such as drug discovery and molecular property prediction, where understanding why a model makes a prediction is essential for scientific validation and regulatory compliance. We do not foresee direct negative societal consequences from this methodological contribution. The computational efficiency gains may democratize access to rigorous explanation methods for practitioners with limited computational resources. As with all explanation techniques, users should remain aware that attributions reflect the model's learned behavior rather than ground-truth causal relationships.

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

Appendix A collects the notation and background needed to read the method and proofs. We first define tensor and tensor-network (TN) notation and the contraction operator used throughout. We then derive the multilinear-extension contraction identity that underpins our deterministic Shapley computation, and show how partial derivatives are computed efficiently via local gate replacement. The remaining sections provide proofs of the main theoretical statements, additional experimental details, and extended discussion/related work.

# A. Preliminaries and notation

This section fixes tensor and TN notation used throughout and states the contraction conventions needed for our multilinear-extension and derivative identities.

### A.1. Tensors and Tensor Networks

Let $\mathcal{T} \in \mathbb{R}^{d_1 \times \cdots \times d_n}$ be an order-$n$ tensor and $x_i \in \mathbb{R}^{d_i}$. The associated multilinear map is

$$g(x_1, \ldots, x_n) := \mathcal{T} \times_1 x_1 \times_2 \cdots \times_n x_n, \tag{16}$$

where $\times_i$ denotes the mode-$i$ tensor–vector contraction, defined as

$$\left(\mathcal{T} \times_i x_i\right)_{j_1, \ldots, j_{i-1}, j_{i+1}, \ldots, j_n} := \sum_{k=1}^{d_i} \mathcal{T}_{j_1, \ldots, j_{i-1}, k, j_{i+1}, \ldots, j_n} (x_i)_k. \tag{17}$$

Successive contractions along all modes yield a scalar.

A tensor network (TN) represents $\mathcal{T}$ implicitly as a contraction of low-order *core tensors* connected by shared indices (bonds). The product of bond dimensions across any bipartition defines the *cut rank*, which controls both expressiveness and contraction complexity. We write $\widehat{\mathcal{T}}(\Theta)$ for the tensor implicitly represented by a TN with parameters $\Theta$.

**Bond dimension.** The bond dimension $\chi$ of each edge controls how much information flows between the connected cores. Larger $\chi$ increases expressivity but also parameter count. The expressivity–cost tradeoff is formalized via the *matricization rank* (Definition 3.1).

**Tensor trains (TT).** A tensor train (TT) is a TN topology in which $\mathcal{T}$ is factorized as

$$\mathcal{T}_{i_1, \ldots, i_n} = \sum_{\alpha_1, \ldots, \alpha_{n-1}} G^{(1)}_{i_1, \alpha_1} G^{(2)}_{\alpha_1, i_2, \alpha_2} \cdots G^{(n)}_{\alpha_{n-1}, i_n}, \tag{18}$$

where $G^{(k)}$ are 3-way cores (except at the ends) and $\alpha_k$ are bond indices of dimension $\chi_k$. The maximal $\chi_k$ determines the TT rank and contraction cost (Oseledets, 2011).

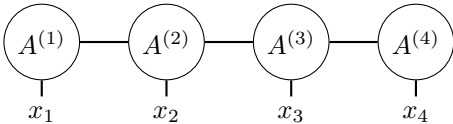

*Figure 8.* Tensor train (TT) representation of a 4-way tensor. Physical indices $x_i$ are shown as dangling legs; internal edges correspond to bond indices.

Evaluating $g(x_1, \ldots, x_n)$ using a TN representation corresponds to contracting all bond indices after attaching the vectors $\{x_i\}$ to the physical legs. We refer to one such evaluation as a *TN contraction*. In TN-SHAP-G, all surrogate evaluations and partial derivatives are computed via TN contractions with appropriately chosen local vectors.

### A.2. Multilinear Extension and tensor view

The multilinear extension admits an equivalent tensor contraction form that will be useful for connecting Shapley computation to tensor-network surrogates. For completeness, a step-by-step derivation of the contraction identity is given in Appendix A.3.

### A.3. Step-by-step derivation of the contraction identity

We derive Equation (8) starting from the Bernoulli-mixture definition of the multilinear extension in Equation (4).

**Step 1 (Start from the multilinear extension).** By definition,

$$\widetilde{\nu}(z) = \sum_{S \subseteq V} \nu(S) \prod_{v \in S} z_v \prod_{v \notin S} (1 - z_v). \tag{19}$$

**Step 2 (Re-index the sum by binary indicators).** Each subset $S \subseteq V$ is in bijection with its indicator vector $s \in \{0,1\}^n$ given by $s_v = \mathbf{1}\{v \in S\}$. Equivalently, for any $s \in \{0,1\}^n$ define $S(s) := \{v \in V : s_v = 1\}$. Using this bijection, we can rewrite (19) as a sum over $s$:

$$\widetilde{\nu}(z) = \sum_{s \in \{0,1\}^n} \nu\big(S(s)\big) \prod_{v=1}^{n} z_v^{s_v} (1 - z_v)^{1 - s_v}. \tag{20}$$

**Step 3 (Introduce the coalition tensor).** Define the coalition tensor $\mathcal{T} \in \mathbb{R}^{2 \times \cdots \times 2}$ by

$$\mathcal{T}_{s_1,\ldots,s_n} := \nu\big(S(s)\big), \qquad s \in \{0,1\}^n.$$

Substituting this into (20) yields

$$\widetilde{\nu}(z) = \sum_{s \in \{0,1\}^n} \mathcal{T}_s \prod_{v=1}^{n} z_v^{s_v} (1 - z_v)^{1 - s_v}. \tag{21}$$

**Step 4 (Define selector vectors).** For each $v \in V$ define

$$\boldsymbol{b}_v(z_v) := \begin{bmatrix} 1 - z_v \\ z_v \end{bmatrix} \in \mathbb{R}^2.$$

Then for $s_v \in \{0,1\}$,

$$\boldsymbol{b}_v(z_v)_{s_v} = (1 - z_v)^{1 - s_v} z_v^{s_v}. \tag{22}$$

Therefore,

$$\prod_{v=1}^{n} z_v^{s_v} (1 - z_v)^{1 - s_v} = \prod_{v=1}^{n} \boldsymbol{b}_v(z_v)_{s_v}.$$

Plugging this into (21) gives

$$\widetilde{\nu}(z) = \sum_{s \in \{0,1\}^n} \mathcal{T}_s \prod_{v=1}^{n} \boldsymbol{b}_v(z_v)_{s_v}. \tag{23}$$

**Step 5 (Recognize the tensor contraction).** Since $\mathcal{T}$ has mode size 2 in each dimension and each $\boldsymbol{b}_v(z_v) \in \mathbb{R}^2$, the multilinear form in (23) is exactly the successive mode-wise contraction of $\mathcal{T}$ with $\{\boldsymbol{b}_v(z_v)\}_{v=1}^{n}$:

$$\sum_{s_1=0}^{1} \cdots \sum_{s_n=0}^{1} \mathcal{T}_{s_1,\ldots,s_n} \prod_{v=1}^{n} \boldsymbol{b}_v(z_v)_{s_v} = \mathcal{T} \times_1 \boldsymbol{b}_1(z_1) \times_2 \cdots \times_n \boldsymbol{b}_n(z_n).$$

This proves the contraction identity (8).

**Exactness of Shapley values.** The multilinear extension $\widetilde{\nu}$ is uniquely determined by the $2^n$ masked evaluations $\nu(S) = f(G, x_S)$. Consequently, any multilinear function $\widehat{\nu}(z)$ that matches these values at the vertices,

$$\widehat{\nu}(\mathbf{1}_S) = \nu(S) \quad \forall S \subseteq V,$$

must coincide with $\widetilde{\nu}$ everywhere on $[0,1]^n$. In particular, Shapley values are given exactly by

$$\phi(v) = \int_0^1 \frac{\partial \widetilde{\nu}}{\partial z_v}(t, \ldots, t) \, dt.$$

This provides a principled foundation for our approach: by learning a multilinear surrogate $\widehat{\nu}$ that fits the masked-input vertices, we recover the unique multilinear extension of the ML game and can compute exact Shapley values via deterministic polynomial integration.

### A.4. Uniqueness of the multilinear extension

*Remark* A.1 (Uniqueness of the multilinear extension). The multilinear extension $\widetilde{\nu}$ is the *unique* multilinear polynomial satisfying $\widetilde{\nu}(\mathbf{1}_S) = \nu(S)$ for all $S \subseteq V$. This follows from the fact that the $2^n$ functions $\{\prod_{v \in S} z_v \prod_{v \notin S}(1 - z_v)\}_{S \subseteq V}$ form a basis for the space of multilinear polynomials in $n$ variables. Consequently, *any* multilinear surrogate $\hat{\nu}$ that perfectly fits the coalition values $\{\nu(S)\}_{S \subseteq V}$ must coincide with $\widetilde{\nu}$, and Shapley values computed via (5) on the surrogate are exact.

**Theorem A.2** (Single-surrogate sufficiency via the multilinear extension). *Let $V = \{1, \ldots, n\}$ and let $\nu : 2^V \to \mathbb{R}$ be a cooperative game with (Owen) multilinear extension $\widetilde{\nu} : [0, 1]^n \to \mathbb{R}$ defined by (39). Then:*
 *(i) **(Uniqueness)** $\widetilde{\nu}$ is the unique multilinear polynomial satisfying $\widetilde{\nu}(\mathbf{1}_S) = \nu(S)$ for all $S \subseteq V$.*
 *(ii) **(Linearity of Shapley)** For each $i \in V$, there exists a linear functional $\mathcal{L}_i$ on multilinear polynomials such that*

$$\phi_i(\nu) = \mathcal{L}_i(\widetilde{\nu}),$$

  *and in particular $\phi_i(\nu)$ admits the equivalent integral representations (40)–(41).*
 *(iii) **(Linearity of interactions)** For any nonempty $T \subseteq V$, $|T| = k \geq 1$, and for any interaction index $\mathcal{I}_T$ that admits a multilinear-extension characterization of the form*

$$\mathcal{I}_T(\nu) = c_k \int_0^1 \frac{\partial^k \widetilde{\nu}}{\partial p_T}(t\mathbf{1}) \, w_k(t) \, dt$$

  *for some scalar $c_k$ and weight function $w_k$ depending only on $k$, there exists a linear functional $\mathcal{L}_T$ such that*

$$\mathcal{I}_T(\nu) = \mathcal{L}_T(\widetilde{\nu}).$$

 *Consequently, for any surrogate multilinear extension $\widehat{\widetilde{\nu}}$ (e.g., represented by a tensor network) we obtain surrogate attributions by substitution:*

$$\widehat{\phi}_i := \mathcal{L}_i(\widehat{\widetilde{\nu}}), \qquad \widehat{\mathcal{I}}_T := \mathcal{L}_T(\widehat{\widetilde{\nu}}),$$

*so a single surrogate $\widehat{\widetilde{\nu}}$ suffices to compute all desired first-order Shapley values and higher-order interaction indices, without retraining separate surrogates per index.*

Proof is in Appendix C.5

**Surrogate-based deterministic attributions.** The discussion above implies a concrete recipe for query-efficient graph explanations. First, we represent the masked game $\nu$ through its multilinear extension $\widetilde{\nu}$, whose Shapley values and low-order interactions reduce to integrals of diagonal partial derivatives (Eq. (5)). Second, since these diagonal derivatives are univariate polynomials of degree at most $n - 1$, they can be recovered from finitely many evaluations via stable interpolation. Finally, we avoid the exponential coalition tensor by learning a structured surrogate $\hat{\nu}$ from a limited query budget, and then compute all required Shapley quantities *deterministically* from this single surrogate without further teacher evaluations. Section 3 instantiates this recipe using graph-aligned tensor networks.

## B. Derivative Computation via Local Gate Replacement

This appendix provides a detailed derivation of how Shapley values and interaction indices are computed from the tensor network surrogate via local gate replacement.

### B.1. Setup and Notation

Recall that the surrogate multilinear extension is given by the tensor contraction

$$\hat{\nu}(z) = \sum_{s \in \{0,1\}^n} \widehat{\mathcal{T}}_s \prod_{v=1}^n [\boldsymbol{b}_v(z_v)]_{s_v}, \tag{24}$$

where $\widehat{\mathcal{T}} \in \mathbb{R}^{2 \times \cdots \times 2}$ is the (implicitly represented) surrogate coalition tensor and the local Bernoulli gate at node $v$ is

$$\boldsymbol{b}_v(z_v) = \begin{bmatrix} 1 - z_v \\ z_v \end{bmatrix}, \qquad [\boldsymbol{b}_v(z_v)]_{s_v} = (1 - z_v)^{1 - s_v} z_v^{s_v}. \tag{25}$$

### B.2. First-Order Derivatives

**Lemma B.1** (Local gate derivative). *For the Bernoulli gate $\boldsymbol{b}_v(z_v) = [1 - z_v, z_v]^\top$, we have*

$$\frac{\partial \boldsymbol{b}_v(z_v)}{\partial z_v} = \begin{bmatrix} -1 \\ 1 \end{bmatrix} =: \boldsymbol{b}'_v, \tag{26}$$

*which is constant (independent of $z_v$).*

*Proof.* Direct computation: $\frac{\partial}{\partial z_v}(1 - z_v) = -1$ and $\frac{\partial}{\partial z_v}(z_v) = 1$. $\square$

**Proposition B.2** (Partial derivative via gate replacement). *The partial derivative of the surrogate with respect to $z_u$ is obtained by replacing the gate at position $u$ with its derivative:*

$$\frac{\partial \hat{\nu}}{\partial z_u}(z) = \sum_{s \in \{0,1\}^n} \widehat{\mathcal{T}}_s \cdot [\boldsymbol{b}'_u]_{s_u} \cdot \prod_{v \neq u} [\boldsymbol{b}_v(z_v)]_{s_v}. \tag{27}$$

*In tensor contraction notation:*

$$\frac{\partial \hat{\nu}}{\partial z_u}(z) = \widehat{\mathcal{T}} \times_1 \boldsymbol{b}_1(z_1) \times_2 \cdots \times_u \boldsymbol{b}'_u \times_{u+1} \cdots \times_n \boldsymbol{b}_n(z_n). \tag{28}$$

*Proof.* Since (24) is a product of terms where only one factor depends on $z_u$, the product rule gives

$$\frac{\partial \hat{\nu}}{\partial z_u}(z) = \sum_{s \in \{0,1\}^n} \widehat{\mathcal{T}}_s \cdot \frac{\partial [\boldsymbol{b}_u(z_u)]_{s_u}}{\partial z_u} \cdot \prod_{v \neq u} [\boldsymbol{b}_v(z_v)]_{s_v} \tag{29}$$

$$= \sum_{s \in \{0,1\}^n} \widehat{\mathcal{T}}_s \cdot [\boldsymbol{b}'_u]_{s_u} \cdot \prod_{v \neq u} [\boldsymbol{b}_v(z_v)]_{s_v}, \tag{30}$$

where we used Lemma B.1 and the fact that

$$\frac{\partial}{\partial z_u} \left[ (1 - z_u)^{1-s_u} z_u^{s_u} \right] = \begin{cases} -1 & \text{if } s_u = 0, \\ +1 & \text{if } s_u = 1, \end{cases} = [\boldsymbol{b}'_u]_{s_u}. \tag{31}$$

$\square$

### B.3. Diagonal Evaluation and Polynomial Structure

**Proposition B.3** (Diagonal derivative is a univariate polynomial). *Define $g_u(t) := \frac{\partial \hat{\nu}}{\partial z_u}(t, \ldots, t)$. Then $g_u$ is a univariate polynomial in $t$ of degree at most $n - 1$:*

$$g_u(t) = \sum_{k=0}^{n-1} a_k^{(u)} t^k. \tag{32}$$

*Proof.* Substituting $z_v = t$ for all $v$ into (27):

$$g_u(t) = \sum_{s \in \{0,1\}^n} \widehat{\mathcal{T}}_s \cdot [\boldsymbol{b}'_u]_{s_u} \cdot \prod_{v \neq u} (1 - t)^{1-s_v} t^{s_v} \tag{33}$$

$$= \sum_{s \in \{0,1\}^n} \widehat{\mathcal{T}}_s \cdot [\boldsymbol{b}'_u]_{s_u} \cdot (1 - t)^{(n-1)-|s_{-u}|} \cdot t^{|s_{-u}|}. \tag{34}$$

Each term $(1 - t)^{(n-1)-|s_{-u}|} t^{|s_{-u}|}$ is a polynomial in $t$. The highest power of $t$ occurs when $|s_{-u}| = n - 1$, giving $t^{n-1}$. Hence $\deg(g_u) \leq n - 1$. $\square$

### B.4. Exact Shapley via Vandermonde interpolation (linked to Theorem 3.8)

Since $g_u(t)$ is a polynomial of degree at most $n - 1$, it is uniquely determined by $n$ evaluations at distinct points.

**Theorem B.4** (Exact Shapley computation). *Let $t_1, \ldots, t_n \in (0, 1)$ be distinct probe points. Define:*

- $y_j := g_u(t_j)$ *for $j = 1, \ldots, n$ (computed via TN contraction with gate replacement),*

- $V \in \mathbb{R}^{n \times n}$ *the Vandermonde matrix with $V_{jk} = t_j^{k-1}$.*

---

**Algorithm 2 TN-SHAP-G** (deterministic O2 edge interactions from a single surrogate)

---

**Require:** trained surrogate $\hat{\nu}(\cdot; \Theta)$ on $G = (V, E)$, probe points $\{t_j\}_{j=1}^m$

**Ensure:** edge interaction indices $\{\hat{\phi}_{uv}\}_{(u,v) \in E}$

 1: **for** $j = 1, \ldots, m$ **do**
 2:    Set all sites to $\boldsymbol{b}_v(t_j)$ and contract TN on $G$ to obtain reusable messages
 3:    **for** each $(u, v) \in E$ **do**
 4:       Compute $g_{uv}(t_j) = \partial_{z_u} \partial_{z_v} \hat{\nu}(t_j \mathbf{1})$
 5:          by replacing $\boldsymbol{b}_u(t_j), \boldsymbol{b}_v(t_j) \mapsto \boldsymbol{b}'_u(t_j), \boldsymbol{b}'_v(t_j)$
 6:    **end for**
 7: **end for**
 8: For each $(u, v)$, interpolate $g_{uv}(t)$ from $\{(t_j, g_{uv}(t_j))\}_{j=1}^m$
 9: Return $\hat{\phi}_{uv} = \int_0^1 g_{uv}(t)\, dt$ for all $(u, v) \in E$

---

*Then the Shapley value is*

$$\widehat{\phi}(u) = \int_0^1 g_u(t)\, dt = \sum_{k=0}^{n-1} \frac{a_k^{(u)}}{k+1}, \tag{35}$$

*where the coefficients $a^{(u)} = (a_0^{(u)}, \ldots, a_{n-1}^{(u)})^\top$ are recovered by solving $V a^{(u)} = y$.*

*Proof.* The Vandermonde system $V a^{(u)} = y$ has a unique solution since the $t_j$ are distinct. Integration of the monomial basis gives the result. $\square$

### B.5. Second-Order Interactions via Double Gate Replacement

**Proposition B.5** (Mixed partial derivatives). *For distinct players $i, j \in V$, the second-order mixed partial derivative is obtained by replacing* both *gates at positions $i$ and $j$:*

$$\frac{\partial^2 \hat{\nu}}{\partial z_i \partial z_j}(z) = \sum_{s \in \{0,1\}^n} \widehat{\mathcal{T}}_s \cdot [\boldsymbol{b}'_i]_{s_i} \cdot [\boldsymbol{b}'_j]_{s_j} \cdot \prod_{v \notin \{i,j\}} [\boldsymbol{b}_v(z_v)]_{s_v}. \tag{36}$$

*Proof.* Apply Proposition B.2 twice: first differentiate with respect to $z_i$ (replacing $\boldsymbol{b}_i \mapsto \boldsymbol{b}'_i$), then differentiate with respect to $z_j$ (replacing $\boldsymbol{b}_j \mapsto \boldsymbol{b}'_j$). Since $\boldsymbol{b}'_i$ and $\boldsymbol{b}'_j$ are constant vectors, the order of differentiation does not matter. $\square$

**Corollary B.6** (Diagonal mixed derivative polynomial). *Define $h_{ij}(t) := \frac{\partial^2 \hat{\nu}}{\partial z_i \partial z_j}(t, \ldots, t)$. Then $h_{ij}$ is a univariate polynomial of degree at most $n - 2$, and the second-order Shapley interaction index is*

$$\widehat{\phi}_{ij} = \int_0^1 h_{ij}(t)\, dt, \tag{37}$$

*computable via Vandermonde interpolation with $n - 1$ probe points.*

### B.6. General $k$-th Order Interactions

**Proposition B.7** ($k$-th order derivative). *For a subset $T \subseteq V$ with $|T| = k$, the $k$-th order mixed partial derivative is*

$$\frac{\partial^k \hat{\nu}}{\partial z_T}(z) = \sum_{s \in \{0,1\}^n} \widehat{\mathcal{T}}_s \cdot \prod_{u \in T} [\boldsymbol{b}'_u]_{s_u} \cdot \prod_{v \notin T} [\boldsymbol{b}_v(z_v)]_{s_v}, \tag{38}$$

*where $\partial z_T := \prod_{u \in T} \partial z_u$. The diagonal restriction is a polynomial of degree at most $n - k$.*

### B.7. Computational Cost

Each evaluation of $g_u(t_j)$ requires one tensor network contraction with modified gates. For $m$ probe points and $n$ players:

- First-order Shapley values: $O(n \cdot m \cdot T_{\mathrm{contr}})$ total cost.

- Second-order interactions (all pairs): $O(n^2 \cdot m \cdot T_{\mathrm{contr}})$ total cost.

- Second-order interactions (edges only): $O(|E| \cdot m \cdot T_{\mathrm{contr}})$ total cost.

Here $T_{\mathrm{contr}}$ is the cost of a single TN contraction, which depends on the graph structure and bond dimension $\chi$.

### B.8. Summary: The Gate Replacement Principle

The key insight enabling efficient exact computation is:

> **Gate Replacement Principle.** To compute $\frac{\partial^k \hat{\nu}}{\partial z_T}$ for any $T \subseteq V$:
>
> 1. For each $u \in T$: replace $\boldsymbol{b}_u(z_u) = [1 - z_u, z_u]^\top$ with $\boldsymbol{b}'_u = [-1, 1]^\top$.
>
> 2. For each $v \notin T$: keep $\boldsymbol{b}_v(z_v)$ unchanged.
>
> 3. Contract the tensor network as usual.
>
> This yields the exact partial derivative at cost equal to one surrogate evaluation.

## C. Proofs

This appendix contains proofs and technical details omitted from the main text for clarity.

### C.1. Cut-rank proof

*Proof.* Cutting the edges in $\delta(A, B)$ exposes a collection of bond indices $\alpha \in \prod_{e \in \delta(A,B)} [\chi_e]$. Contracting all cores on the $A$ side yields functions $f_\alpha(s_A)$, and contracting all cores on the $B$ side yields functions $g_\alpha(s_B)$, such that

$$\mathcal{T}_s = \sum_\alpha f_\alpha(s_A)\, g_\alpha(s_B).$$

Thus the $(A|B)$-matricization factors as a sum of at most $\prod_{e \in \delta(A,B)} \chi_e$ rank-one terms, proving (11). $\square$

**Proof of Corollary 3.3.** This is a direct consequence of Theorem 3.2. If all edges in $\delta(A, B)$ have uniform bond dimension $\chi$, then $\prod_{e \in \delta(A,B)} \chi_e = \chi^{|\delta(A,B)|}$. Substituting into (11) yields $\mathrm{rank}_{A|B}(\mathcal{T}) \leq \chi^{|\delta(A,B)|}$. $\square$

### C.2. Proof and formal statement of Theorem C.1 (TT-rank inflation)

**Theorem C.1** (TT-rank inflation from tensor-ring structure). *Let $\mathcal{T} \in \mathbb{R}^{n_1 \times \cdots \times n_d}$ admit a tensor-ring (TR) decomposition with cores $\mathcal{G}^{(j)} \in \mathbb{R}^{r_j \times n_j \times r_{j+1}}$ for $j \in [d]$, with cyclic ranks $r_{d+1} := r_1$. For any $k \in [d-1]$, let $T_{(k)} \in \mathbb{R}^{N_{\leq k} \times N_{>k}}$ denote the matricization with $N_{\leq k} = \prod_{j=1}^k n_j$ and $N_{>k} = \prod_{j=k+1}^d n_j$. Then:*
 *(i) **(Upper bound)** $\mathrm{rank}(T_{(k)}) \leq r_1 r_{k+1}$.*
 *(ii) **(Generic tightness)** If $N_{\leq k} \geq r_1 r_{k+1}$ and $N_{>k} \geq r_1 r_{k+1}$, then for a generic choice of TR cores, $\mathrm{rank}(T_{(k)}) = r_1 r_{k+1}$.*
 *(iii) **(TT lower bound)** Any tensor-train (TT) representation of $\mathcal{T}$ with the same mode ordering must have TT rank at bond $k$ satisfying $\rho_k \geq r_1 r_{k+1}$.*
 *(iv) **(TT approximation error bound)** For any TT tensor $\mathcal{T}_{\mathrm{TT}}$ with bond rank $\rho_k$ at cut $k$,*

$$\|\mathcal{T} - \mathcal{T}_{\mathrm{TT}}\|_F \geq \Big( \sum_{i > \rho_k} \sigma_i^2(T_{(k)}) \Big)^{1/2},$$

*where $\sigma_i(T_{(k)})$ denotes the $i$-th singular value of the matricization $T_{(k)}$.*
*In particular, if $r_1 = \cdots = r_d = \chi$, then generically any TT representation requires TT ranks $\Omega(\chi^2)$.*

**Proof sketch.** Fix $k \in [d-1]$. Contract the first $k$ TR cores into a left subchain and the remaining $d - k$ cores into a right subchain, yielding a factorization $T_{(k)} = \mathbf{L}\mathbf{R}$ with $\mathbf{L} \in \mathbb{R}^{N_{\leq k} \times (r_1 r_{k+1})}$ and $\mathbf{R} \in \mathbb{R}^{(r_1 r_{k+1}) \times N_{>k}}$ (see, e.g., (Zhao et al., 2016)). This implies $\mathrm{rank}(T_{(k)}) \leq r_1 r_{k+1}$. Under the stated dimension conditions, $\mathbf{L}$ and $\mathbf{R}$ are generically full rank, so equality holds. Finally, TT rank at bond $k$ equals $\mathrm{rank}(T_{(k)})$ for the same ordering, proving the lower bound.

For (iv), note that reshaping preserves Frobenius norm, so $\|\mathcal{T} - \mathcal{T}_{\mathrm{TT}}\|_F = \|T_{(k)} - T_{\mathrm{TT},(k)}\|_F$. Since TT structure forces $\mathrm{rank}(T_{\mathrm{TT},(k)}) \leq \rho_k$, the Eckart–Young theorem gives $\|T_{(k)} - A\|_F \geq (\sum_{i > \rho_k} \sigma_i^2(T_{(k)}))^{1/2}$ for any rank-$\rho_k$ matrix $A$, with equality attained by the truncated SVD.

**Proof of Corollary 3.5** Consider a TT/MPS surrogate with node ordering $\pi$ and TT-ranks $\rho_1, \ldots, \rho_{n-1}$. Any chain cut at position $k$ along $\pi$ induces a bipartition $(A, B) = (\{\pi(1), \ldots, \pi(k)\}, \{\pi(k+1), \ldots, \pi(n)\})$ of the player set. The TT representation has a single bond of dimension $\rho_k$ at this cut. By Theorem 3.2, $\rho_k \geq \mathrm{rank}_{A|B}(\mathcal{T})$.

For GNN-induced games, the interaction complexity across $(A, B)$ is governed by the walk index of the cut boundary (Razin et al., 2023): $\mathrm{rank}_{A|B}(\mathcal{T})$ can be large for cuts whose boundary admits many length-$(L-1)$ walks crossing the

partition. Thus, for orderings $\pi$ that place highly interacting nodes on opposite sides of a chain cut, the required TT-rank $\rho_k$ can be large.

In contrast, a graph-aligned TN distributes bond capacity along all graph edges simultaneously. It is ordering-invariant by construction: every graph separator is directly controlled by the bond dimensions of the edges it cuts, without requiring that the separator be expressible as a single chain cut. $\square$

### C.3. Proof sketch of Shapley stability

*Proof.* The bound follows by writing $\phi_T$ as an average of $2^{|T|}$-term discrete differences, each involving evaluations of $\nu$ and $\hat{\nu}$ that differ by at most $\varepsilon$. This follows from the linearity of $\phi$ and the fact that each marginal contribution involves two game evaluations, while each $k$-th order discrete difference involves $2^k$ game evaluations; see Heidari et al. (2025). $\square$

### C.4. Proof of Theorem 3.6

*Proof sketch.* (Khavari & Rabusseau, 2021) bound the pseudo-dimension of TN hypothesis classes by $\mathrm{Pdim}(\mathcal{H}_G) \leq 2N_G \log(12|V|)$ (Khavari & Rabusseau, 2021, Thm. 2). Standard learning-theory bounds for bounded regression classes controlled by pseudo-dimension yield an excess-risk rate $\tilde{O}\big(B^2(\mathrm{Pdim}(\mathcal{H}_G) + \log(1/\delta))/M\big)$, which becomes $\tilde{O}\big(B^2(N_G + \log(1/\delta))/M\big)$ after substitution. Finally, if $\deg_{\max}(G) \leq \Delta$ and $\dim(e) \leq \chi$, then for each $v$, $\prod_{e \in E_v} \dim(e) \leq \chi^{\deg(v)} \leq \chi^\Delta$, hence $N_G \leq \sum_{v \in V} \chi^\Delta = |V|\chi^\Delta$. $\square$

**Proof of Theorem 3.6.** Let $\mathcal{H}_G$ be the TN hypothesis class on topology $G$ with parameter count $N_G$. By Theorem 2 of Khavari & Rabusseau (2021), its pseudo-dimension satisfies $\mathrm{Pdim}(\mathcal{H}_G) \leq 2N_G \log(12|V|)$; write $d := \mathrm{Pdim}(\mathcal{H}_G)$.

Assume $\nu(S) \in [-B, B]$ and consider squared-loss risk $\mathcal{E}(h) := \mathbb{E}_{S \sim \mathcal{D}}[(h(S) - \nu(S))^2]$, estimated by ERM $\hat{\nu}$ over $M$ i.i.d. samples. Since the loss is bounded by $O(B^2)$, standard uniform-convergence bounds for real-valued function classes controlled by pseudo-dimension give, with probability $\geq 1 - \delta$,

$$\mathcal{E}(\hat{\nu}) - \inf_{h \in \mathcal{H}_G} \mathcal{E}(h) = \tilde{O}\left( B^2 \sqrt{\frac{d + \log(1/\delta)}{M}} \right) = \tilde{O}\left( B^2 \sqrt{\frac{N_G + \log(1/\delta)}{M}} \right),$$

where $\tilde{O}$ absorbs the $\log(12|V|)$ and $\log M$ factors.

For the parameter count: if $\deg_{\max}(G) \leq \Delta$ and $\dim(e) \leq \chi$, then each core $\mathbf{A}^{(u)}$ has $2 \prod_{e \in E_v} \dim(e) \leq 2\chi^{\deg(u)} \leq 2\chi^\Delta$ entries, so $N_G \leq 2|V|\chi^\Delta$. Setting the excess risk $\leq \varepsilon$ therefore requires

$$M = \tilde{O}\left( \frac{B^4\big(|V|\chi^\Delta + \log(1/\delta)\big)}{\varepsilon^2} \right),$$

which is near-linear in $|V|$ for fixed $\chi, \Delta, B$. $\square$

### C.5. Proof of Single-Surrogate Sufficiency via the Multilinear Extension (Theorem A.2)

*Proof.* Fix a player set $V = \{1, \ldots, n\}$ and a game $\nu : 2^V \to \mathbb{R}$. Its (Owen) multilinear extension is the polynomial

$$\widetilde{\nu}(p) = \sum_{S \subseteq V} \nu(S) \prod_{i \in S} p_i \prod_{i \notin S} (1 - p_i), \qquad p \in [0, 1]^n. \tag{39}$$

This polynomial is *unique*: since the $2^n$ functions $\big\{ \prod_{i \in S} p_i \prod_{i \notin S} (1 - p_i) \big\}_{S \subseteq V}$ form a basis of the space of multilinear polynomials in $(p_1, \ldots, p_n)$, the coefficients $\nu(S)$ are uniquely determined, hence $\widetilde{\nu}$ is uniquely determined by $\nu$.

We now show that all standard attribution quantities obtained from $\nu$ (Shapley values and higher-order interaction indices) can be written as *linear functionals* of $\widetilde{\nu}$.

Let $e_i$ be the $i$-th standard basis vector. The Shapley value admits the (Owen) integral representation

$$\phi_i(\nu) = \int_0^1 \Big( \widetilde{\nu}(t\mathbf{1} + (1-t)e_i) - \widetilde{\nu}(t\mathbf{1}) \Big) \, dt, \tag{40}$$

equivalently (by the fundamental theorem of calculus),

$$\phi_i(\nu) = \int_0^1 \frac{\partial \widetilde{\nu}}{\partial p_i}(t\mathbf{1}) \, dt. \tag{41}$$

Both (40) and (41) are linear in $\widetilde{\nu}$: evaluation $\widetilde{\nu} \mapsto \widetilde{\nu}(p)$, partial differentiation $\widetilde{\nu} \mapsto \partial_{p_i}\widetilde{\nu}$, and integration are linear operators. Hence $\phi_i(\nu) = \mathcal{L}_i(\widetilde{\nu})$ for a linear functional $\mathcal{L}_i$.

For any $T \subseteq V$ with $|T| = k \geq 1$, a broad class of $k$-th order interaction indices (including Shapley–Taylor / Shapley interaction families) can be written as averaged $k$-th mixed partial derivatives of the same multilinear extension:

$$\mathcal{I}_T(\nu) \;=\; c_k \int_0^1 \frac{\partial^k \widetilde{\nu}}{\partial p_T}(t\mathbf{1})\, w_k(t)\, dt, \tag{42}$$

for some scalar constant $c_k$ and weight function $w_k$ depending only on $k$ (and on the chosen interaction definition), where $\partial p_T := \prod_{i \in T} \partial p_i$. Again, (42) is linear in $\widetilde{\nu}$ because mixed partial differentiation and integration are linear. Therefore $\mathcal{I}_T(\nu) = \mathcal{L}_T(\widetilde{\nu})$ for a linear functional $\mathcal{L}_T$.

Since each desired attribution quantity is a linear functional of the *same* unique $\widetilde{\nu}$, any surrogate $\hat{\nu}$ that approximates $\widetilde{\nu}$ (e.g., a trained tensor-network surrogate) can be plugged into these linear functionals to obtain corresponding approximations to all first-order Shapley values and higher-order interactions, without retraining a separate surrogate for each index. □

### C.6. Runtime proof (Section 3.3)
*Proof.* Let $(T, \{X_t\}_{t \in T})$ be a tree decomposition of $G$ with width $\tau$, so each bag satisfies $|X_t| \leq \tau + 1$.

**Step 1: Cluster tensor construction.** For each bag $X_t$, define the cluster tensor $\mathcal{C}_t \in \mathbb{R}^{\chi^{|X_t|}}$ by contracting all core tensors $\{\mathbf{A}^{(u)}\}_{u \in X_t}$ whose indices are internal to the bag, while leaving bond indices connecting to other bags as open legs. Each cluster tensor has at most $\tau + 1$ open bond indices.

**Step 2: Message passing on the tree.** The tree decomposition $T$ induces a tree structure on clusters. Run two-pass message passing:

- **Upward pass:** Root $T$ arbitrarily. For each cluster $t$ with children $c_1, \ldots, c_k$, compute the message to parent:

$$\mu_{t \to \text{parent}(t)} = \text{Contract}\big(\mathcal{C}_t(t\mathbf{1}), \mu_{c_1 \to t}, \ldots, \mu_{c_k \to t}\big),$$

  marginalizing over indices not shared with the parent bag.

- **Downward pass:** Compute messages from parents to children analogously.

Each message contraction involves tensors with at most $\tau + 1$ indices of dimension $\chi$, costing $O(\chi^{\tau+1})$ per edge of $T$. Since $|T| = O(|V|)$, the total message-passing cost is $O(|V|\chi^{\tau+1})$.

**Step 3: Environment computation.** For any node $u \in V$, let $t(u)$ denote a bag containing $u$. The environment $\mathcal{E}_u$ (the full contraction with site $u$ removed) is computed by:

1. Contracting all incoming messages to bag $t(u)$: cost $O(\chi^{\tau+1})$.

2. Removing the contribution of $\mathbf{A}^{(u)}$ from the cluster tensor.

With messages precomputed, each environment costs $O(\chi^{\tau+1})$.

**Step 4: Derivative computation.** Given $\mathcal{E}_u$, the first-order derivative is:

$$\partial_{z_u} \hat{\nu}(t\mathbf{1}) = \mathcal{E}_u \times_{\text{phys}} \mathbf{b}'_u,$$

where $\mathbf{b}'_u = [-1, 1]^\top$. This costs $O(\chi^\tau)$ per node.

For edges $(u, v) \in E$, if $u$ and $v$ lie in the same bag (which can always be arranged for edges), the edge environment $\mathcal{E}_{uv}$ is constructed similarly by removing both sites from the cluster, and:

$$\partial_{z_u} \partial_{z_v} \hat{\nu}(t\mathbf{1}) = \mathcal{E}_{uv} \times (\mathbf{b}'_u \otimes \mathbf{b}'_v).$$

**Step 5: Total complexity.**

- Message passing (once per probe point): $O(|V|\chi^{\tau+1})$

- All $|V|$ node environments and derivatives: $O(|V|\chi^{\tau+1})$

- All $|E|$ edge environments and derivatives: $O(|E|\chi^{\tau+1}) = O(|V|\chi^{\tau+1})$

Summing and multiplying by $m$ probe points yields the stated bound. □

Together, Theorems 3.8 and 3.6 show that TN-SHAP-G computes Shapley values and interaction indices exactly on the learned surrogate, with runtime and query complexity determined by graph structure. All attribution quantities are obtained from a single surrogate without additional teacher queries.

### C.7. Approximation floor from limited cut capacity

**Lemma C.2** (Approximation floor via truncated SVD). *Let $\mathcal{T} \in \mathbb{R}^{2 \times \cdots \times 2}$ be the coalition tensor and fix any bipartition $(A, B)$ of the node set $V$. Let $\widehat{\mathcal{T}}$ be any surrogate tensor such that its matricization across this cut satisfies $\mathrm{rank}(\widehat{\mathcal{T}}_{A|B}) \leq r$. Then the best achievable approximation error is lower-bounded by the truncated SVD residual of the true matricization:*

$$\|\mathcal{T} - \widehat{\mathcal{T}}\|_F \;\geq\; \|\mathcal{T}_{A|B} - \widehat{\mathcal{T}}_{A|B}\|_F \;\geq\; \|\mathcal{T}_{A|B} - (\mathcal{T}_{A|B})_r\|_F \;=\; \Big( \sum_{k>r} \sigma_k(\mathcal{T}_{A|B})^2 \Big)^{1/2}, \tag{43}$$

*where $(\mathcal{T}_{A|B})_r$ denotes the best rank-$r$ approximation of $\mathcal{T}_{A|B}$, obtained by truncating its singular value decomposition.*

*Proof.* Matricization is a linear reshaping, so $\|\mathcal{T} - \widehat{\mathcal{T}}\|_F = \|\mathcal{T}_{A|B} - \widehat{\mathcal{T}}_{A|B}\|_F$ for any fixed bipartition $(A, B)$. Since $\mathrm{rank}(\widehat{\mathcal{T}}_{A|B}) \leq r$, the matrix $\widehat{\mathcal{T}}_{A|B}$ is a rank-$\leq r$ approximation of $\mathcal{T}_{A|B}$. By the Eckart–Young theorem, the minimum Frobenius error among rank-$r$ approximations of $\mathcal{T}_{A|B}$ is achieved by truncating its SVD, and equals $\|\mathcal{T}_{A|B} - (\mathcal{T}_{A|B})_r\|_F$. This yields (43). $\qquad\square$

### C.8. Exact Shapley via polynomial interpolation (Theorem 3.8)

**Theorem C.3** (Exact Shapley computation via polynomial interpolation). *Let $\hat{\nu} : [0,1]^n \to \mathbb{R}$ be a multilinear surrogate. For any player $u \in \mathcal{P}$, define*

$$g_u(t) \;:=\; \frac{\partial \hat{\nu}}{\partial z_u}(t, \ldots, t).$$

*Then $g_u$ is a univariate polynomial of degree at most $n - 1$, and the Shapley value*

$$\widehat{\phi}(u) = \int_0^1 g_u(t)\, dt$$

*is exactly recoverable from evaluations of $g_u$ at $n$ distinct points $t_1, \ldots, t_n \in (0, 1)$ via polynomial interpolation (equivalently, by solving a Vandermonde linear system).*

*Proof.* Because $\hat{\nu}$ is multilinear, it admits the unique multilinear expansion

$$\hat{\nu}(z) = \sum_{S \subseteq \mathcal{P}} a_S \prod_{i \in S} z_i, \qquad z \in \mathbb{R}^n, \tag{44}$$

for some coefficients $\{a_S\}_{S \subseteq \mathcal{P}}$ (with $a_\emptyset$ the constant term). In particular, each variable appears in each monomial with exponent either 0 or 1.

Fix $u \in \mathcal{P}$. Differentiating (44) with respect to $z_u$ gives

$$\frac{\partial \hat{\nu}}{\partial z_u}(z) = \sum_{S \subseteq \mathcal{P}:\, u \in S} a_S \prod_{i \in S \setminus \{u\}} z_i. \tag{45}$$

This is a multilinear polynomial in the remaining $n - 1$ variables $\{z_i\}_{i \neq u}$, and every monomial in (45) has total degree $|S| - 1 \leq n - 1$.

Now restrict to the diagonal $z = t\mathbf{1}$, i.e., set $z_i = t$ for all $i \in \mathcal{P}$. Plugging into (45) yields

$$g_u(t) = \frac{\partial \hat{\nu}}{\partial z_u}(t\mathbf{1}) = \sum_{S \subseteq \mathcal{P}:\, u \in S} a_S\, t^{|S|-1}. \tag{46}$$

Equation (46) is a univariate polynomial in $t$ whose highest possible power is $t^{n-1}$, hence $\deg(g_u) \leq n - 1$.

Write $g_u(t) = \sum_{k=0}^{n-1} c_k t^k$ for coefficients $c_0, \ldots, c_{n-1}$. Choose $n$ distinct points $t_1, \ldots, t_n \in (0, 1)$ and evaluate $g_u$ at them:

$$g_u(t_j) = \sum_{k=0}^{n-1} c_k t_j^k, \qquad j = 1, \ldots, n.$$

Let $V \in \mathbb{R}^{n \times n}$ be the Vandermonde matrix $V_{j,k+1} = t_j^k$ (for $j = 1, \ldots, n$ and $k = 0, \ldots, n-1$), let $c = (c_0, \ldots, c_{n-1})^\top$, and let $y = (g_u(t_1), \ldots, g_u(t_n))^\top$. Then the above relations are exactly the linear system

$$Vc = y. \tag{47}$$

Since the $t_j$ are distinct, the Vandermonde determinant satisfies $\det(V) = \prod_{1 \le i < j \le n}(t_j - t_i) \ne 0$, so $V$ is invertible and (6) has a unique solution. Therefore the coefficient vector $c$ (and thus $g_u$) is uniquely and exactly determined by the $n$ evaluations $\{g_u(t_j)\}_{j=1}^n$.

Finally, integrate the recovered polynomial:

$$\widehat{\phi}(u) = \int_0^1 g_u(t)\, dt = \sum_{k=0}^{n-1} \frac{c_k}{k+1}.$$

Because $c$ is recovered exactly from $y$ via the invertible Vandermonde system, the value of $\widehat{\phi}(u)$ computed by the above expression is exact. $\qquad\square$

## D. Coalition sampling and dataset construction

**Mixture sampling distribution.**    We sample coalitions from a mixture distribution

$$T \sim \mathcal{D} := \lambda_{\text{size}}\mathcal{D}_{\text{size}} + \lambda_{\text{conn}}\mathcal{D}_{\text{conn}} + \lambda_{\text{O2}}\mathcal{D}_{\text{O2}}, \qquad \lambda_{\text{size}} + \lambda_{\text{conn}} + \lambda_{\text{O2}} = 1, \tag{48}$$

where each component targets a complementary regime of the coalition space.

**Size-stratified component.**    In $\mathcal{D}_{\text{size}}$, we first sample a coalition size $k \in \{0, \dots, n\}$ from a distribution $p(k)$ and then sample $T$ uniformly among coalitions of size $k$:

$$k \sim p(\cdot), \qquad T \,\big|\, |T| = k \sim \text{Unif}\{T \subseteq V : |T| = k\}.$$

We use $p(k) = \text{Unif}\{0, \dots, n\}$ by default to ensure balanced coverage across hypercube layers.

**Connected-coalition component.**    In $\mathcal{D}_{\text{conn}}$, we sample connected coalitions of a target size $k$ to emphasize graph-local perturbations. We first sample $k \sim p(k)$, then: (i) draw a seed $u \sim \text{Unif}(V)$; (ii) iteratively grow a set $T \leftarrow \{u\}$ by adding a node from its boundary $\partial T$ until $|T| = k$. We implement the growth step by choosing the next node uniformly at random from $\partial T$ (random BFS-style expansion), which produces connected induced subgraphs.

**Second-order-aware component.**    In $\mathcal{D}_{\text{O2}}$, we oversample coalitions that directly constrain first- and second-order effects. Specifically, with probability $1/2$ we draw a singleton and with probability $1/2$ we draw a pair:

$$T = \begin{cases} \{u\}, & u \sim \text{Unif}(V), \\ \{u, v\}, & (u, v) \sim q(\cdot), \end{cases}$$

where $q$ may be uniform over all unordered pairs or biased toward edges (e.g., $q(u, v) \propto \mathbb{1}[(u, v) \in E]$) to target local interactions.

**Train/validation/test protocol.**    We construct a labeled dataset $\{(T_j, \nu(T_j))\}_{j=1}^M$ by independently sampling coalitions from (48). We then split coalitions into disjoint subsets $\mathcal{T}_{\text{train}}$, $\mathcal{T}_{\text{val}}$, and $\mathcal{T}_{\text{test}}$. Validation is used for early stopping and selecting hyperparameters (including bond dimension $\chi$ and mixture weights $\lambda$), while $\mathcal{T}_{\text{test}}$ is held out for final reporting.

TN-SHAP-G computes Shapley values (and interaction indices) exactly from the learned surrogate $\hat{\nu}$. Thus, the dominant source of error is the surrogate generalization error on the masked game. We therefore report held-out $R^2$ on coalitions as a direct measure of surrogate fidelity, and use it as a principled proxy for attribution accuracy.

### D.1. Connection to Harsanyi dividends (Möbius coefficients)

Recall the Harsanyi dividend (Möbius) expansion of a multilinear game $\hat{\nu} : [0, 1]^n \to \mathbb{R}$:

$$\hat{\nu}(z) = \sum_{S \subseteq V} w(S) \prod_{v \in S} z_v, \tag{49}$$

where $\{w(S)\}_{S \subseteq V}$ are the Möbius coefficients.

Fix a player $u \in V$ and define the diagonal partial derivative

$$g_u(t) \; := \; \frac{\partial \hat{\nu}}{\partial z_u}(t, \dots, t).$$

Differentiating (49) yields

$$\frac{\partial \hat{\nu}}{\partial z_u}(z) = \sum_{S \ni u} w(S) \prod_{v \in S \setminus \{u\}} z_v,$$

and evaluating on the diagonal $z = (t, \ldots, t)$ gives the univariate polynomial

$$g_u(t) = \sum_{S \ni u} w(S)\, t^{|S|-1}. \tag{50}$$

Writing $g_u(t) = \sum_{r=0}^{n-1} a_r t^r$, we obtain the coefficient identity

$$a_r = \sum_{\substack{S \ni u \\ |S| = r+1}} w(S), \qquad r = 0, \ldots, n-1. \tag{51}$$

Thus, the diagonal-derivative interpolation coefficients $\{a_r\}$ computed in Section 3.3 provide a size-wise aggregation of interaction terms involving $u$: $a_0$ captures singleton contributions, while larger $r$ collect higher-order cooperative structure.

### D.2. Statistical limits of sampling-based Shapley in low-dispersion games

We formalize why Monte Carlo estimation of Shapley values becomes ill-conditioned when the game exhibits low dispersion under coalition sampling.

Let $u$ be a fixed player and define the marginal contribution random variable

$$\Delta_u(S) = v(S \cup \{u\}) - v(S), \qquad S \sim \pi_u,$$

where $\pi_u$ is the coalition distribution induced by random permutations or KernelSHAP weighting. The Shapley value is

$$\phi(u) = \mathbb{E}[\Delta_u(S)].$$

Assume observations of $v(\cdot)$ are corrupted by additive zero-mean noise:

$$\widetilde{v}(S) = v(S) + \xi_S, \qquad \mathbb{E}[\xi_S] = 0, \quad \mathrm{Var}(\xi_S) = \sigma^2_{\text{noise}},$$

with independent noise across coalitions. Then each sampled marginal contribution is observed as

$$\widetilde{\Delta}_u(S) = \widetilde{v}(S \cup \{u\}) - \widetilde{v}(S) = \Delta_u(S) + (\xi_{S \cup \{u\}} - \xi_S),$$

with noise variance $2\sigma^2_{\text{noise}}$.

**Proposition D.1** (Relative sample complexity lower bound). *Let $\widehat{\phi}(u) = \frac{1}{K} \sum_{k=1}^{K} \widetilde{\Delta}_u(S_k)$ be the standard Monte Carlo estimator from $K$ i.i.d. samples $S_k \sim \pi_u$. Then*

$$\mathrm{Var}[\widehat{\phi}(u)] = \frac{\mathrm{Var}[\Delta_u(S)] + 2\sigma^2_{noise}}{K}.$$

*To achieve relative accuracy $|\widehat{\phi}(u) - \phi(u)| \leq \varepsilon |\phi(u)|$ with constant probability, it is necessary that*

$$K \gtrsim \frac{\mathrm{Var}[\Delta_u(S)] + 2\sigma^2_{noise}}{\varepsilon^2 \phi(u)^2}. \tag{52}$$

*In particular, if $|\phi(u)| \ll \sigma_{noise}$, the required number of samples diverges as $K = \Omega(\sigma^2_{noise}/\phi(u)^2)$.*

*Proof.* Since the samples are i.i.d.,

$$\mathrm{Var}[\widehat{\phi}(u)] = \frac{1}{K}\mathrm{Var}[\widetilde{\Delta}_u(S)] = \frac{\mathrm{Var}[\Delta_u(S)] + 2\sigma^2_{\text{noise}}}{K}.$$

By Chebyshev's inequality (or a Gaussian approximation), achieving $|\widehat{\phi}(u) - \phi(u)| \leq \varepsilon |\phi(u)|$ with constant probability requires

$$\mathrm{Var}[\widehat{\phi}(u)] \lesssim \varepsilon^2 \phi(u)^2,$$

which yields (52). $\square$

In low-dispersion regimes, $\Delta_u(S)$ has small variance, but the Shapley values $\phi(u)$ typically shrink at a comparable or faster rate due to redundancy in the game. As a result, the ratio $\mathrm{Var}[\Delta_u(S)]/\phi(u)^2$ remains large, and any nonzero noise floor $\sigma_{\text{noise}}$ makes relative estimation statistically ill-conditioned. This explains why permutation-based and KernelSHAP estimators fail to recover meaningful attributions even when $v(S)$ appears smooth.

### D.3. Interaction indices

We define the higher-order interaction indices used throughout this work.

#### D.3.1. SECOND-ORDER SHAPLEY INTERACTION INDEX

For a cooperative game $\nu : 2^V \to \mathbb{R}$, the *Shapley interaction index* between players $i, j \in V$ quantifies their joint contribution beyond their individual effects. Following Grabisch & Roubens (1999), the second-order interaction index is

$$\phi_{ij}(\nu) = \sum_{S \subseteq V \setminus \{i,j\}} \frac{|S|! \, (n - |S| - 2)!}{(n - 1)!} \, \Delta_{ij} \nu(S), \tag{53}$$

where $\Delta_{ij} \nu(S)$ is the discrete second-order difference

$$\Delta_{ij} \nu(S) = \nu(S \cup \{i, j\}) - \nu(S \cup \{i\}) - \nu(S \cup \{j\}) + \nu(S).$$

A positive interaction index $\phi_{ij} > 0$ indicates complementarity (synergy), while a negative value indicates redundancy.

#### D.3.2. MULTILINEAR EXTENSION CHARACTERIZATION

Interaction indices admit a multilinear-extension representation. In particular,

$$\phi_{ij}(\nu) = \int_0^1 \frac{\partial^2 \widetilde{\nu}}{\partial z_i \partial z_j}(t, \dots, t) \, dt.$$

Since $\widetilde{\nu}$ is multilinear, the mixed partial derivative is multilinear in the remaining $n - 2$ variables. Restricting to $z = t\mathbf{1}$ yields a univariate polynomial of degree at most $n - 2$, which can be exactly integrated via the same Vandermonde interpolation procedure used for first-order values.

#### D.3.3. HIGHER-ORDER INTERACTIONS

For a subset $T \subseteq V$ with $|T| = k$, the $k$-th order interaction index generalizes naturally:

$$\phi_T(\nu) = \sum_{S \subseteq V \setminus T} \frac{|S|! \, (n - |S| - k)!}{(n - k + 1)!} \, \Delta_T \nu(S), \tag{54}$$

where $\Delta_T \nu(S)$ is the $k$-th order discrete difference. The multilinear-extension characterization extends to

$$\phi_T(\nu) = c_k \int_0^1 \frac{\partial^k \widetilde{\nu}}{\partial z_T}(t\mathbf{1}) \, w_k(t) \, dt,$$

where $c_k$ and $w_k$ depend only on the interaction definition.

## E. Experimental set up

**Reproducibility.** All experiments use only public benchmark datasets and standard GNN architectures. When reporting runtimes, we report wallclock measured on a single commodity GPU and include the full timing breakdown (sampling / training / attribution) for transparency.

Unless stated otherwise, we explain a fixed test instance $(G, X)$ using the node-masking game in (2). Players are nodes $V(G) = \{1, \dots, n\}$. Given a coalition $S \subseteq V(G)$, we construct a masked feature matrix $X^{(S)}$ by replacing features of excluded nodes with a baseline vector. Our default baseline is the *mean-feature baseline*:

$$x_i^{(S)} = \begin{cases} x_i & i \in S, \\ \bar{x} := \frac{1}{n} \sum_{j=1}^n x_j & i \notin S. \end{cases} \tag{55}$$

The game value is the black-box model score on the masked graph:

$$\nu(S) := f\big(G, X^{(S)}\big), \tag{56}$$

where $f : \mathcal{G} \to \mathbb{R}$ is a trained graph-level predictor. Unless otherwise stated, $f$ outputs the *logit of the predicted class* (where the predicted class is determined on the unmasked instance $(G, X)$).

*Table 3.* **Query budget for** $> 0.95$ **cosine.**

| Method | O1 | O2 | Myerson |
|---|---|---|---|
| Exact | $2^n$ | $2^n$ | $2^n$ |
| Permutation | 500 | — | — |
| SHAP-IQ | 5000 | 5000 | — |
| **TN-SHAP-G** | **50** | **$+0^\dagger$** | **$+0^\dagger$** |

$^\dagger$Same surrogate; no additional queries.

For each explained graph, we train a surrogate $\hat{\nu}$ to approximate $\nu$ from $M$ teacher queries $\{(S_m, \nu(S_m))\}_{m=1}^{M}$ by minimizing MSE:

$$\min_{\hat{\nu} \in \mathcal{H}} \frac{1}{M} \sum_{m=1}^{M} \left( \hat{\nu}(S_m) - \nu(S_m) \right)^2. \tag{57}$$

We report surrogate fidelity using held-out coalition $R^2$ (train/val/test splits) computed on the coalition-value regression problem.

Given a trained surrogate, we compute (i) O1 Shapley values and (ii) O2 interaction indices via the multilinear extension and deterministic quadrature / interpolation, as described in Section 4. For small graphs ($n \le 20$), we compute ground-truth attributions via exhaustive enumeration over all $2^n$ coalitions and report attribution fidelity using cosine similarity (and, where applicable, MAE and rank correlation).

We use two main budget regimes: (i) *low-data* training with $M = 10n$ coalitions per graph (used for structure ablations and stress tests), and (ii) *accuracy-focused* training with $M$ scaling between $20n$ and $40n$ depending on $n$ (used for large-graph scalability experiments). Each sample set includes $S = \emptyset$ and $S = V$.

**Tensor contractions and implementation.** All tensor network contractions are performed with `cotengra` (Gray & Kourtis, 2021). Unless otherwise stated, we fix the random seed for coalition sampling and surrogate initialization and report mean±std over graphs (and, when relevant, across repeated runs).

## F. Extended experimental details

This section provides additional experimental details, implementation specifics, and extended analyses supporting the results presented in Section 4. We describe dataset preprocessing, model configurations, surrogate training procedures, evaluation protocols, and scalability settings that were omitted from the main text due to space constraints, and include supplementary figures and tables referenced therein.

### F.1. Correctness on Small Graphs (extended)

We validate **TN-SHAP-G** against exact enumeration on graphs where exhaustive computation of all $2^n$ coalitions is feasible ($n \le 20$). This provides ground-truth O1 Shapley values and O2 interactions.

**Data and protocol.** We draw small graphs from Mutagenicity (Debnath et al., 1991) by filtering to instances with $n \le 20$. For each explained graph: (i) we compute $\nu(S)$ for all $S \subseteq V$ by enumerating $2^n$ coalitions; (ii) we train a TN surrogate from $M$ teacher queries (budget as specified in the experiment); (iii) we recover O1 and O2 from the trained surrogate without additional teacher queries.

We report cosine similarity between surrogate-derived and exact attributions, averaged over nodes (O1) and over pairs (O2), along with mean±std over graphs. We also report surrogate fidelity (test $R^2$) and relate it to attribution accuracy.

For each graph, the TN surrogate is trained once using $\mathcal{O}(nm)$ teacher queries to cover all players and interaction probes (up to order 2). After training, Shapley values and interactions are computed without additional teacher queries. Table 1 compares all methods. We omit O2 results for GraphSVX since the method is designed for first-order Shapley attributions and does not support higher-order interaction indices.

### F.2. Query Efficiency and Baseline Comparison

We compare TN-SHAP-G against permutation sampling and SHAP-IQ (Fumagalli et al., 2024), evaluating on small graphs ($n \le 20$) from Mutagenicity where exact enumeration provides ground truth.

Figure 9 shows accuracy distributions at budget 500. TN-SHAP-G exhibits $3-5\times$ smaller interquartile range than sampling methods, as Shapley extraction is deterministic once the surrogate is trained.

We compare against two representative model-agnostic estimators:

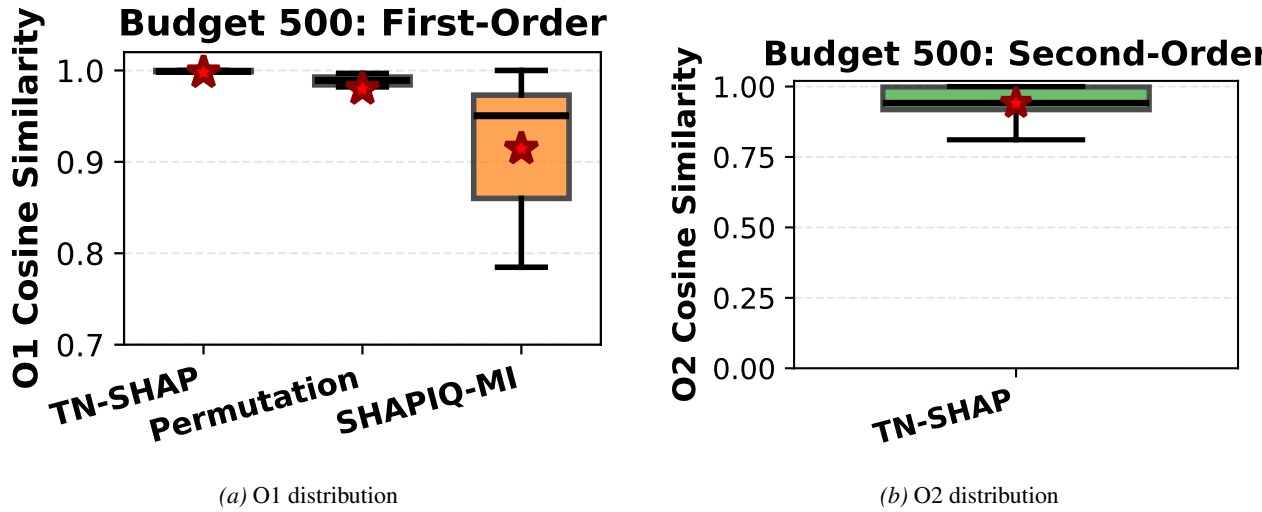

*(a)* O1 distribution          *(b)* O2 distribution

*Figure 9.* **Accuracy at budget 500.** TN-SHAP-G has higher median and $3-5\times$ lower variance than sampling methods.

- **Permutation sampling** (Castro et al., 2009) for O1 (and, when applicable, for O2 via the standard inclusion–exclusion marginal on sampled permutations).
- **SHAP-IQ** (Fumagalli et al., 2024) as a recent interaction-focused estimator.

All methods are evaluated on small graphs ($n \leq 20$) from Mutagenicity where exact enumeration provides ground truth. We report cosine similarity versus teacher-query budget. Budgets are matched as closely as possible across methods (counting a single $\nu(S)$ evaluation as one teacher query).

Table 3 summarizes query requirements across methods and orders.

### F.3. Qualitative Explanations on a Graph Transformer

We present a qualitative case study demonstrating that TN-SHAP-G applies beyond message-passing GNNs by explaining predictions of a graph transformer (Graphormer-style) model on Mutagenicity. The goal is to illustrate the method's generality and the qualitative structure of the resulting node- and interaction-level attributions, rather than to optimize predictive performance.

#### F.3.1. BLACK-BOX MODEL

We use a simplified Graphormer-style architecture (Ying et al., 2021) with linear node embeddings ($d$=128), a learnable `[CLS]` token for graph-level readout, and shortest-path distance (SPD) embeddings added as an attention bias. The model consists of 4 transformer layers with 8 attention heads, feedforward width 512, and dropout 0.1. Classification is performed by an MLP applied to the `[CLS]` representation. The model is trained on Mutagenicity using an 80/10/10 train/validation/test split with AdamW (lr $10^{-4}$, weight decay $10^{-2}$) and cosine learning-rate decay over 200 epochs.

#### F.3.2. GAME DEFINITION AND SURROGATE TRAINING

We define a soft node-masking game compatible with the multilinear extension by applying continuous masks $m_i \in [0, 1]$ at the input embedding level:

$$h_i^{(m)} = m_i \cdot \text{Embed}(x_i).$$

The game value is the logit of the predicted class on the masked graph. For each selected molecule, we train a TN surrogate from 500 stratified coalition samples, using bond dimension $\chi = 8$ and early stopping. Only surrogates with validation $R^2 \geq 0.90$ are retained for attribution.

#### F.3.3. ATTRIBUTION COMPUTATION AND VISUALIZATION

From the trained surrogate, we compute O1 Shapley values via one-dimensional quadrature and O2 interaction indices via two-dimensional quadrature, as described in Appendix D.3. Attributions are visualized using three complementary views: (i) bar plots of node Shapley values, (ii) heatmaps of pairwise interactions among the most influential nodes, and (iii) 2D molecular renderings colored by node attribution (generated with RDKit (Landrum et al., 2021)).

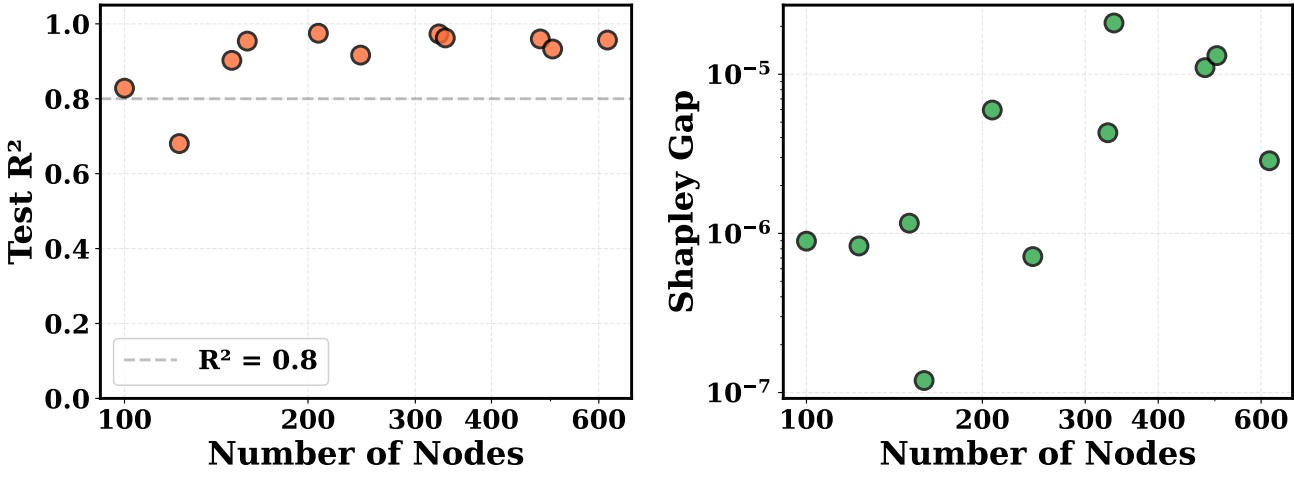

*(a)* Test $R^2$ vs Number of Nodes (100–600 range)     *(b)* Shapley Gap vs Number of Nodes (log scale, $10^{-7}$ to $10^{-5}$)

*Figure 10.* **Scalability on PROTEINS.** T(a) Surrogate test $R^2$ remains above 0.8 across graph sizes up to 600 nodes. (b) Shapley efficiency gap $|\sum_i \phi_i - (\nu(V) - \nu(\emptyset))|$ stays small ($10^{-7}$–$10^{-5}$), confirming numerical stability of the deterministic recovery.

*Table 4.* GA-TN advantage over the best TT variant (O2 correlation with ground truth).

| #params | GA-TN O2 | Best TT O2 | Gap |
|---|---|---|---|
| $\sim$4100 | $0.852 \pm 0.110$ | $0.806 \pm 0.102$ | **+4.6%** |
| $\sim$6900 | $0.931 \pm 0.061$ | $0.778 \pm 0.149$ | **+15.3%** |
| $\sim$8900 | $0.954 \pm 0.046$ | $0.878 \pm 0.081$ | **+7.6%** |

### F.3.4. QUALITATIVE OBSERVATIONS

Across representative Mutagenicity molecules, high-magnitude node attributions and strong interactions concentrate on chemically meaningful functional groups (e.g., nitro and aromatic motifs), consistent with known mutagenicity mechanisms. Compared to GIN-based explanations (Appendix F.3), Graphormer explanations tend to be more spatially distributed and exhibit more long-range interactions, reflecting the model's global self-attention and structural bias. This suggests that TN-SHAP-G captures model-specific reasoning encoded in the induced game, rather than producing architecture-agnostic saliency patterns. Figure 11 shows representative TN-SHAP-G explanations on two Mutagenicity molecules. The left panels report node-level Shapley values, the center panels show pairwise interaction strengths, and the right panels visualize node attributions on the molecular graph.

### F.4. Structure ablation details

We compare: (i) **graph-aligned TN surrogates (GA-TN)** whose factorization follows the input graph topology, and (ii) **tensor-train (TT/MPS)** surrogates which impose a chain topology over an ordering of nodes.

To control for capacity, we match surrogates by total parameter count. For TT, we evaluate multiple node orderings: `default`, `bfs`, and `random`. Where default ordering refers to Simplified Molecular Input Line Entry System (SMILES) (Weininger, 1988) and BFS stands for Breadth First Search We observe that SMILE ordering outperform random and BFS in all mutagenicity experiments using tensor trains. All surrogates are trained under the same low-data budget of $10n$ coalition queries per graph.

GA-TN consistently yields higher test $R^2$ and higher attribution recovery (O1 and O2). TT surrogates additionally exhibit sensitivity to node ordering, reflected in broader variance bands in Fig. 12.

### F.5. Experimental setup (full hyperparameters)

**Datasets and splits.** We evaluate on four graph classification benchmarks spanning molecular and protein domains. Table 6 summarizes dataset statistics. When training black-box predictors, we use standard train/val/test splits and select the best checkpoint by validation performance.

For Mutagenicity and PROTEINS, we use a 3-layer Graph Isomorphism Network (GIN) (Xu et al., 2019) with 64 hidden channels, BatchNorm, dropout $p$=0.5, and global add pooling. Each GIN layer uses a 2-layer MLP: Linear→ReLU→Linear.

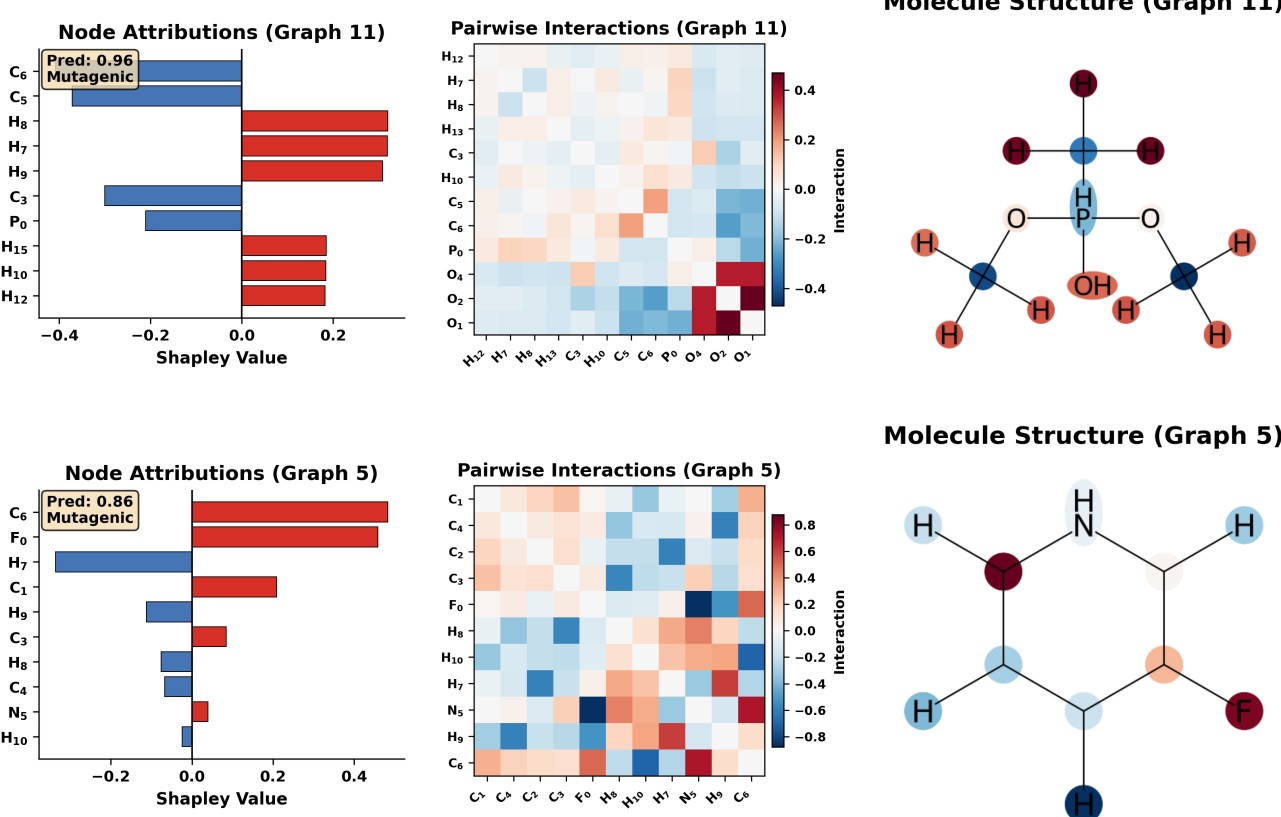

*Figure 11.* TN-SHAP-G explanations on Mutagenicity. **Top:** Organophosphorus compound (prediction 0.96). **Bottom:** Fluoroaniline (prediction 0.86). **Left:** node-level Shapley values. **Center:** pairwise Shapley interaction matrix. **Right:** molecular graph colored by node attribution.

For Benzene, we follow common practice in GraphXAI-style settings and use a standard GCN (Kipf, 2016) (with pretrained checkpoints when available).

For graphs in the enumeration and moderate regimes (typically $n \leq 50$ unless stated otherwise), the TN surrogate mirrors the graph topology: each node $i$ corresponds to a tensor $T_i$ with one physical dimension and bond dimension $\chi \in \{2, 4, 6, 8\}$ connecting to neighbors. We train by sampling $M$ coalitions stratified by size (details below), minimizing MSE with AdamW (lr $10^{-3}$, weight decay $10^{-5}$, 500–2000 epochs). For larger graphs ($n > 30$) in this regime, we cap the number of coalitions to 512–2048 to limit compute.

We use stratified sampling over coalition sizes to cover both sparse and dense coalitions. Concretely, we sample coalition sizes $|S|$ uniformly from $\{0, 1, \ldots, n\}$ (or from a truncated range when evaluating graph-restricted games), then sample $S$ uniformly among subsets of that size. We always include $S = \emptyset$ and $S = V$ and de-duplicate coalitions.

**Deterministic Shapley / interaction recovery.** Given a trained surrogate, we compute:
- **O1 Shapley** via 1D quadrature on the multilinear extension, using $m = 16$ quadrature nodes (Chebyshev or Gauss–Legendre; both give stable results in our setting).
- **O2 interactions** analogously via mixed partials of the multilinear extension, with the same quadrature resolution and edge-restriction when evaluating interaction maps over graphs.

These procedures incur *no additional teacher queries*: they only call the surrogate.

Across experiments, we report: (i) train/val/test $R^2$ for surrogate fidelity, (ii) cosine similarity to ground truth (enumeration) for O1/O2 correctness when tractable, (iii) Shapley efficiency gap $|\sum_{i=1}^{n} \phi_i - (\nu(V) - \nu(\emptyset))|$, and (iv) wallclock time decomposed into sampling, training, and attribution phases.

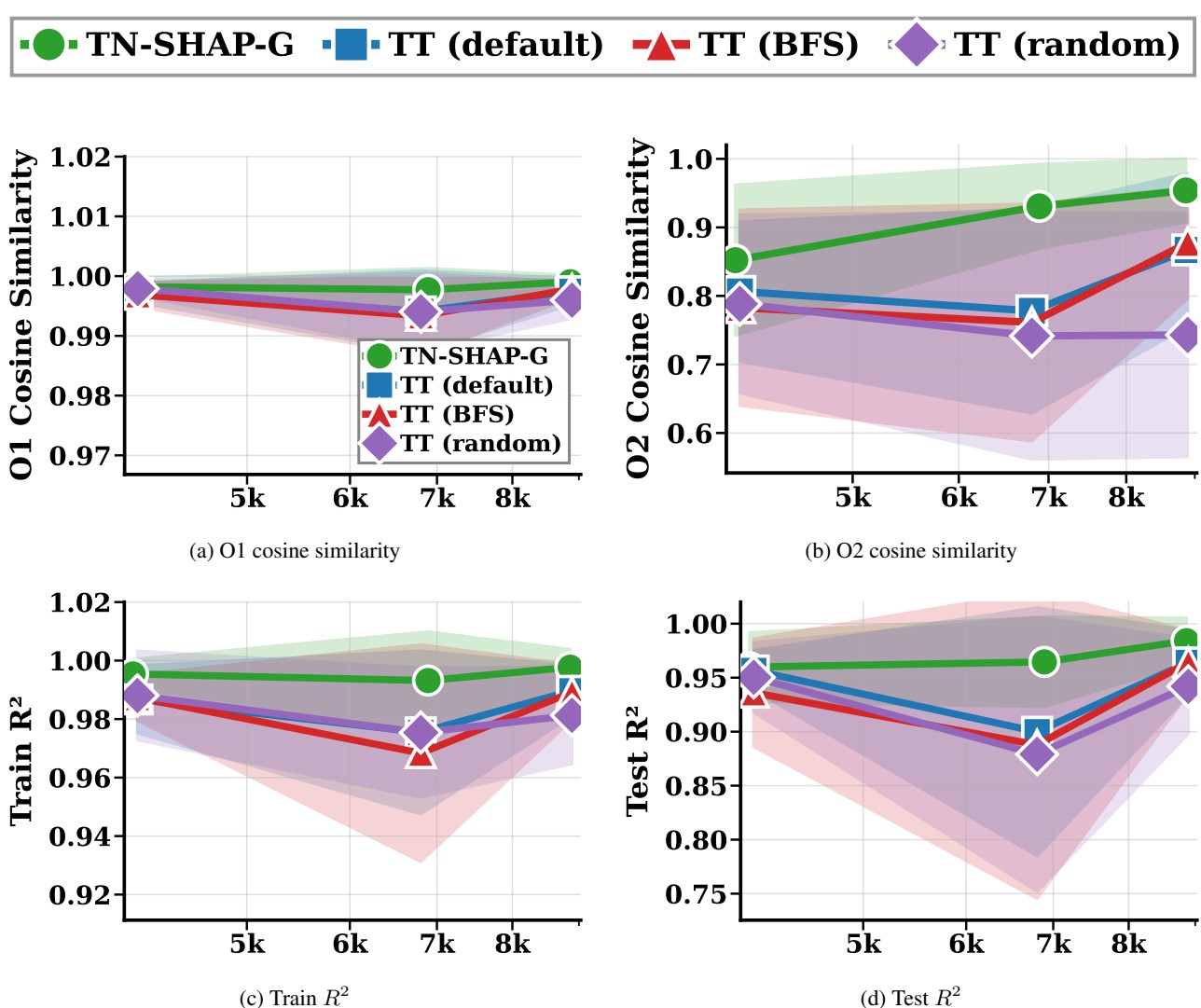

*Figure 12.* **Structure ablation with parameter-matched surrogates.** GA-TN (TN-SHAP-G) consistently achieves higher surrogate fidelity (Train/Test $R^2$) and higher attribution recovery (O1/O2 cosine similarity) than TT surrogates across different orderings at comparable parameter budgets.

### F.6. PROTEINS Dataset Experiments

We evaluate on the PROTEINS dataset from the TU Dortmund graph benchmark collection (Borgwardt et al., 2005; Debnath et al., 1991). The dataset contains 1,113 protein structure graphs where nodes represent secondary structure elements (SSEs) and edges connect neighboring SSEs in 3D space or along the amino-acid sequence. Each node has 3 features encoding SSE attributes. The task is binary graph classification, with graph sizes ranging from 4 to 620 nodes.

**Black-box model.** We use a GIN (Xu et al., 2019) classifier with 3 GIN layers, 64 hidden channels, BatchNorm, ReLU activations, dropout $p = 0.5$, and global add pooling. Each GIN layer uses a 2-layer MLP: Linear($d_{\text{in}}, 64$) $\rightarrow$ ReLU $\rightarrow$ Linear($64, 64$).

### F.7. Scaling

To evaluate scalability across graph sizes, we stratify graphs into 5 size bins: [100–149], [150–199], [200–299], [300–399], and [400–620] nodes. We select 2 graphs per bin (10 total), chosen to be evenly spaced within each bin. Figure 10 shows that surrogate fidelity ($R^2 > 0.8$) and Shapley efficiency (gap $< 10^{-5}$) are maintained as graph size increases to 600 nodes.

For large graphs, we use the coarsening-based surrogate described in Section F.11. We partition nodes into $K$ supernodes, compute $d$ region channels, and train a TN over the coarsened graph. Hyperparameters are adapted by graph size to manage

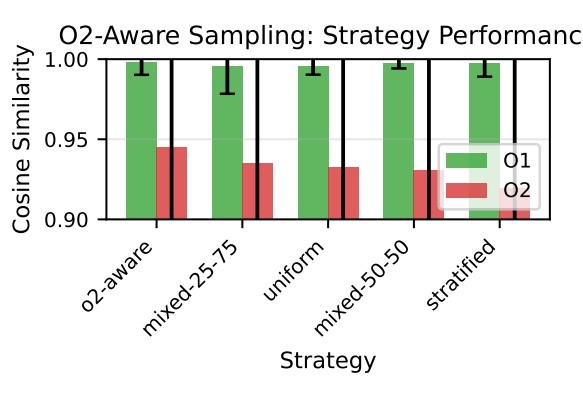

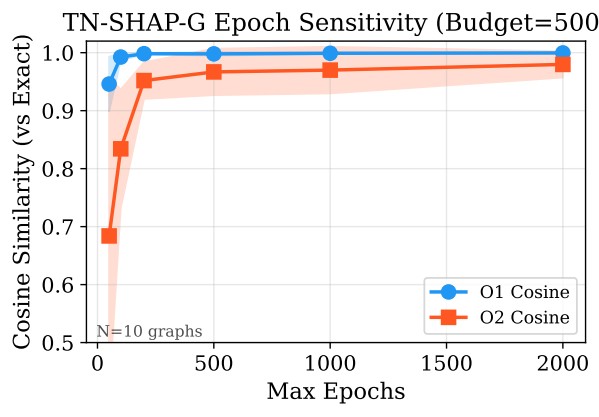

*(a)* Sampling strategy comparison

*(b)* Epoch sensitivity

*Figure 13.* **Ablation studies.** (a) O2-aware sampling achieves highest accuracy for both O1 and O2 indices by preferentially sampling intermediate-size coalitions. (b) O1 Shapley values saturate at $\sim$100 epochs while O2 interactions require $\sim$500 epochs (budget: 500 samples, $N$=10 graphs).

*Table 5.* TN surrogate hyperparameters by graph size for PROTEINS scalability experiments.

| Parameter | $n \leq 150$ | $150 < n \leq 250$ | $250 < n \leq 400$ | $n > 400$ |
|---|---|---|---|---|
| Supernodes $K$ | 18 | 16 | 14 | 12 |
| Channels $d$ | 8 | 6 | 4 | 4 |
| Bond dimension $\chi$ | 6 | 4 | 4 | 3 |
| Epochs | 200 | 250 | 300 | 400 |
| Batch size | 256 | 128 | 64 | 32 |

computational cost:

All configurations use AdamW with learning rate $10^{-3}$ and weight decay $10^{-4}$, a OneCycleLR schedule (20% warmup, cosine annealing), early stopping with patience 150, and gradient clipping (max norm 1.0).

We use stratified coalition sampling with budgets scaling with graph size: $20n$ samples for $n \leq 150$, $30n$ for $150 < n \leq 300$, and $40n$ for $n > 300$. We split sampled coalitions into 70%/15%/15% train/val/test. Each split contains both $\emptyset$ and $V$.

We apply node masking using the mean-feature baseline in (55). The value $\nu(S)$ is the black-box logit for the predicted class (argmax on $(G, X)$).

We compute O1 Shapley values from the surrogate using 16-point Gauss–Legendre quadrature:

$$\phi_i = \int_0^1 \Big[\hat{\nu}(m_i{=}1, m_{-i}{=}t) - \hat{\nu}(m_i{=}0, m_{-i}{=}t)\Big] dt \approx \sum_{k=1}^{16} w_k \Big[\hat{\nu}(m_i{=}1, m_{-i}{=}t_k) - \hat{\nu}(m_i{=}0, m_{-i}{=}t_k)\Big]. \quad (58)$$

For each graph we report: (i) train/val/test $R^2$, (ii) Shapley efficiency gap $|\sum_i \phi_i - (\nu(V) - \nu(\emptyset))|$, and (iii) wallclock time decomposed into sampling / training / Shapley computation. Results are summarized in the scaling plots in the main text (Fig. 7) and the extended scaling section below.

### F.8. Datasets summury

We evaluate on four graph classification datasets spanning molecular chemistry and protein structure. Table 6 summarizes dataset statistics (node ranges and feature dimensions are dataset-intrinsic). Treewidth is computed via a standard min-degree heuristic on the unweighted graph.

#### F.8.1. MUTAGENICITY DATASET

Mutagenicity (Debnath et al., 1991) contains 4,337 molecular graphs (atoms as nodes, bonds as edges) labeled by mutagenicity.

Mutagenicity contains both small graphs (enabling exact enumeration) and mid-sized graphs (enabling scaling tests), making it suitable for correctness and efficiency evaluation.

*Table 6.* Dataset statistics. Treewidth is computed via the min-degree heuristic.

| Dataset | #Graphs | Node Range | Avg. Nodes | Node Feat. | Classes | Treewidth |
|---|---|---|---|---|---|---|
| Mutagenicity | 4,337 | 4–417 | $\sim$30 | 14 | 2 | 1–6 |
| Benzene | 12,000 | 9–36 | $\sim$17 | 14 | 2 | 2–4 |
| PROTEINS | 1,113 | 4–620 | $\sim$39 | 3 | 2 | 2–8 |

### F.8.2. BENZENE DATASET

Benzene from GraphXAI (Sanchez-Lengeling et al., 2020) contains 12,000 synthetic molecular graphs. The task is to detect whether a benzene ring substructure is present.

### F.8.3. PROTEINS DATASET

PROTEINS (Dobson & Doig, 2003) contains 1,113 protein graphs with node features encoding SSE attributes and binary labels.

The wide node range up to 620 allows stress-testing the scalability of the surrogate and the deterministic Shapley recovery pipeline.

### F.9. Black-Box Models

**GIN for Mutagenicity/PROTEINS.** We use a 3-layer GIN (Xu et al., 2019) with 64 hidden channels, BatchNorm, dropout $p$=0.5, and global add pooling. Each layer uses a 2-layer MLP: Linear→ReLU→Linear.

**GCN for Benzene.** Following common practice (Agarwal et al., 2023), we use a 3-layer GCN (Kipf, 2016) with 128 hidden channels and a standard graph-level readout.

### F.10. Experimental Protocol

We use mean-feature node masking (55) and define the game value by the predicted-class logit.

**Reported metrics.** We report:
- **Surrogate quality**: Train/val/test $R^2$
- **Correctness**: Cosine similarity (and optional MAE / rank correlation) vs. ground truth when tractable
- **Efficiency gap**: $|\sum_i \phi_i - (\nu(V) - \nu(\emptyset))|$
- **Runtime**: Wallclock time for sampling, training, and inference (separately)

### F.11. Scalability Experiments

We evaluate computational scalability on graphs of increasing size, measuring surrogate fit quality and wallclock time.

#### F.11.1. COARSENING-BASED SCALING FOR LARGE GRAPHS

For large graphs ($n \gtrsim 100$), direct training of a full graph-aligned TN can become expensive. We therefore introduce a coarsening-based approach that replaces $n$ nodes by $K \ll n$ regions (supernodes), trains a surrogate over the coarsened graph, and then maps region-level attributions back to nodes.

**Method overview.**
1. **Graph coarsening:** partition nodes into $K$ regions via spectral clustering on the adjacency matrix.
2. **Region channels:** compute region-channel features from node masks:

$$z_{r,k} = \sum_{i \in \text{region}_r} A_{r,i,k}\, m_i, \quad r \in [K],\ k \in [d]. \tag{59}$$

3. **Coarsened TN:** train a TN surrogate on the coarsened graph $G_c$ mapping $\{z_{r,:}\}_{r=1}^K$ to $\mathbb{R}$.
   After computing Shapley values for region-channel features, we map them back to nodes:

$$\phi_i = \sum_{k=1}^d A_{r(i),i,k}\, \phi_z[r(i), k], \tag{60}$$

where $r(i)$ denotes the region containing node $i$.

#### F.11.2. PROTEINS SCALABILITY EXPERIMENT

We apply the above scalable surrogate to PROTEINS (Section F.6). The experiment configuration is summarized in Table 7, and the resulting training / attribution times appear in Fig. 7.

*Table 7.* PROTEINS scalability experiment configuration (coarsening-based surrogate).

| Bin | Node Range | Supernodes $K$ | Channels $d$ | Bond $\chi$ | Budget |
|---|---|---|---|---|---|
| Small | 100–149 | 18 | 8 | 6 | $20n$ |
| Medium | 150–199 | 16 | 6 | 4 | $30n$ |
| Large | 200–299 | 14 | 4 | 4 | $30n$ |
| Very Large | 300–399 | 14 | 4 | 4 | $40n$ |
| Huge | 400–620 | 12 | 4 | 3 | $40n$ |

### F.11.3. SECOND-ORDER (O2) SCALING EXPERIMENT

To keep O2 tractable on larger graphs, we restrict O2 computations to adjacent node pairs (edges), reducing the number of interaction terms from $\binom{n}{2}$ to $|E|$.

### F.11.4. TIMING ANALYSIS AND COMPARISON

We report timing in three parts: (i) coalition sampling / teacher querying, (ii) surrogate training, and (iii) surrogate-based attribution. This ensures fair comparison to baselines whose costs scale mainly with teacher queries.

### F.12. Comparison with GraphSHAP-IQ

We compare TN-SHAP-G with GraphSHAP-IQ (Muschalik et al., 2025), a recent method for computing exact Shapley interaction indices on graph neural networks. GraphSHAP-IQ leverages the locality of GNN computations by restricting interaction computations to coalitions within each node's $L$-hop receptive field, where $L$ is the depth of the GNN.

Under this formulation, GraphSHAP-IQ computes exact Shapley values by enumerating all coalitions within each node's local neighborhood. For a node whose $L$-hop neighborhood contains $k$ nodes, this requires $\mathcal{O}(2^k)$ model evaluations. Aggregated across all nodes, the total computational budget scales as $\mathcal{O}(n \cdot 2^{k_{\max}})$, where $k_{\max}$ denotes the maximum neighborhood size over all nodes. While this is substantially more efficient than naïve $\mathcal{O}(2^n)$ enumeration, the exponential dependence on $k_{\max}$ remains a fundamental bottleneck, particularly for graphs with dense local structure.

Table 8 reports the empirical relationship between maximum neighborhood size and computational budget on Mutagenicity graphs. The observed budget closely follows the expected $\mathcal{O}(n \cdot 2^{k_{\max}})$ scaling. As $k_{\max}$ increases beyond 15–17 nodes, runtime and memory usage grow rapidly, leading to practical infeasibility.

*Table 8.* GraphSHAP-IQ budget versus maximum neighborhood size on Mutagenicity. The budget grows exponentially with $k_{\max}$, consistent with $\mathcal{O}(n \cdot 2^{k_{\max}})$ scaling.

| Nodes | $k_{\max}$ | Budget | $2^{k_{\max}}$ | Runtime (s) |
|---|---|---|---|---|
| 8 | 8 | 256 | 256 | 1.4 |
| 12 | 12 | 4,096 | 4,096 | 3.6 |
| 16 | 10 | 2,607 | 1,024 | 1.7 |
| 36 | 13 | 17,037 | 8,192 | 12.7 |
| 41 | 16 | 86,319 | 65,536 | 171.1 |
| 47 | 17 | 286,782 | 131,072 | 928.2 |

In practice, GraphSHAP-IQ becomes infeasible when graphs exhibit dense $L$-hop neighborhoods. Graphs with $k_{\max} > 16$ require hundreds of thousands of model evaluations, often exceeding 15 minutes per instance, and the associated Möbius representations demand $\mathcal{O}(2^{k_{\max}})$ memory per node. As a result, graphs with more than $\sim 60$ nodes are skipped entirely in our experiments.

TN-SHAP-G avoids this exponential dependence by learning a global tensor-network surrogate of the coalition value function. The training cost scales as $\mathcal{O}(M \cdot n)$, where $M$ is the number of sampled coalitions (typically 500–2000), and is independent of neighborhood density. Once trained, the same surrogate supports deterministic recovery of all Shapley values and interaction indices without additional model queries.

On the PROTEINS dataset, which contains graphs with up to 620 nodes, GraphSHAP-IQ is applicable to fewer than 30% of graphs due to neighborhood-size constraints, whereas TN-SHAP-G successfully produces explanations for all graphs using a coarsening-based surrogate. This highlights the complementary regimes of the two methods: GraphSHAP-IQ provides exact attributions on small, sparse graphs, while TN-SHAP-G leverages a compositional multilinear surrogate to enable scalable attribution. By operating through the multilinear extension, TN-SHAP-G admits provable guarantees that relate Shapley and interaction accuracy to the quality of the learned surrogate, providing a principled accuracy–scalability

trade-off.

### F.13. Baseline Sensitivity

To evaluate how the choice of masking baseline affects the quality of recovered Shapley values and interaction indices. We run TN-SHAP-G with three different baselines assigned to masked (absent) nodes:

- **Mean** (paper default): $\mathbf{x}_{\text{base}} = \frac{1}{n}\sum_v \mathbf{x}_v$.
- **Zero**: $\mathbf{x}_{\text{base}} = \mathbf{0}$.
- **Ones**: $\mathbf{x}_{\text{base}} = \mathbf{1}$.

For each baseline we select 30 small graphs ($n \leq 20$) from MUTAGENICITY and train a graph-aligned TN with bond dimension 8 for 3 000 epochs. Ground truth is obtained by exhaustive enumeration of all $2^n$ coalitions under the matching baseline.

*Table 9.* Baseline sensitivity on MUTAGENICITY ($n \leq 20$, 30 graphs per baseline).

| Baseline | O1 Cosine ↑ | O2 Cosine ↑ | Test $R^2$ ↑ | Game Std |
|---|---|---|---|---|
| Mean (default) | $0.997 \pm 0.003$ | $0.868 \pm 0.107$ | $0.953 \pm 0.047$ | $0.20 \pm 0.19$ |
| Zero | $0.976 \pm 0.089$ | $0.878 \pm 0.096$ | $0.919 \pm 0.109$ | $0.24 \pm 0.19$ |
| Ones | $0.998 \pm 0.003$ | $0.719 \pm 0.110$ | $0.991 \pm 0.008$ | $6.45 \pm 1.96$ |

O1 Shapley values are robust to the choice of baseline: all three achieve cosine similarity $\geq 0.976$. For O2, the mean and zero baselines perform comparably ($\sim 0.87$), while the ones baseline yields lower O2 (0.72) due to the substantially larger game variance it induces (game std $\approx 6.5$ vs. $\approx 0.2$). The mean baseline provides the best overall balance. TN-SHAP-G requires no modification to work with any baseline—only the coalition value function changes.

### F.14. Node-Removal vs. Feature-Masking Semantics

We compare two coalition game definitions on the same 15 graphs ($n \leq 14$, MUTAGENICITY) with the same O2-aware Hamming neighbor sampling ($\sim 480$ training masks per graph):

- **Feature masking** (paper default): absent nodes receive mean-baseline features; graph structure unchanged. Bond dimension = 8.
- **Node removal**: absent nodes are deleted; the GNN receives the induced subgraph $G[S]$. Bond dimension = 12.

Ground truth is obtained by exact enumeration under the matching game semantics.

*Table 10.* Coalition game semantics comparison on MUTAGENICITY ($n \leq 14$, 15 graphs, O2-aware sampling).

| Game Semantics | O1 Cosine ↑ | O2 Cosine ↑ | Test $R^2$ ↑ | Bond Dim |
|---|---|---|---|---|
| Feature masking | $0.9999 \pm 0.0002$ | $0.991 \pm 0.017$ | $0.997 \pm 0.005$ | 8 |
| Node removal | $0.989 \pm 0.008$ | $0.886 \pm 0.138$ | $0.996 \pm 0.005$ | 12 |

TN-SHAP-G successfully learns the node-removal game: O1 cosine 0.989, test $R^2 = 0.996$. This confirms that the TN multilinear extension framework generalizes beyond feature masking to structurally defined games where nodes are removed entirely. O2 is lower for node removal (0.886 vs. 0.991) because removing nodes changes the GNN message-passing topology, making the value function less smooth.

### F.15. Myerson Game Semantics

To demonstrate that TN-SHAP-G is not tied to a single game semantics, we evaluate the compute time and fidelity of the graph-restricted Myerson game $v^G(S) = \sum_{C \in \text{CC}(G[S])} v(C)$, where $\text{CC}(G[S])$ denotes the connected components of the induced subgraph. Crucially, this requires no architectural changes: the same TN surrogate is trained on coalition values and reused under the Myerson semantics.

We evaluate the compute time and cosine similarity between the Myerson values computed on the surrogate with those from the exact computation, on the first 20 BENZENE graphs; exact Myerson comparison is feasible for 3 graphs with $n < 15$.

TN-SHAP-G achieves mean O1 cosine 0.965 and O2 cosine 0.969 for Myerson values, with TN surrogate $R^2 = 0.994$. This supports the claim that TN-SHAP-G is a flexible cooperative-game approximation framework, not limited to standard Shapley semantics.

*Table 11.* TN-SHAP-G with Myerson game semantics on BENZENE (exact-feasible subset, $n < 15$).

| Graph | $n$ | TN $R^2$ | O1 Cos | O2 Cos | TN time (s) | Exact time (s) |
|---|---|---|---|---|---|---|
| 2 | 14 | 0.991 | 0.988 | 0.998 | 0.76 | 168.9 |
| 6 | 8 | 0.993 | 0.912 | 0.923 | 0.05 | 1.6 |
| 8 | 13 | 0.998 | 0.996 | 0.987 | 0.66 | 89.3 |
| **Mean** | | **0.994** | **0.965** | **0.969** | | |

### F.16. Continuous-Feature Synthetic Benchmark

To confirm that TN-SHAP-G is not limited to categorical features, we evaluate on a synthetic BA-plus-house motif regression task (MAGE generator) with *continuous* node features. Each graph has 11 nodes with a 5-node house motif determining the regression target. We train a GIN regressor (test $R^2 = 0.988$) and evaluate TN-SHAP-G on 11 held-out test graphs with exact O1 Shapley enumeration over all $2^{11} = 2048$ coalitions.

*Table 12.* Exact O1 Shapley comparison on MAGE synthetic benchmark (11 graphs, continuous features).

| Semantics | O1 Cosine ↑ | O1 $R^2$ ↑ | TN Queries / Exact Queries |
|---|---|---|---|
| Feature masking | $0.994 \pm 0.004$ | 0.980 | 115 / 2048 |
| Node removal | $0.993 \pm 0.005$ | 0.860 | 115 / 2048 |

TN-SHAP-G recovers exact first-order Shapley values with cosine similarity $> 0.99$ under both semantics, using only 115 training queries ($18\times$ fewer than exact enumeration). This confirms that categorical features are not a limitation of the method.

### F.17. Structure-Aware Sampling Ablation

We evaluate how graph-structure-aware coalition sampling affects higher-order interaction recovery. We restrict O2 evaluation to graph edges and O3 to graph triangles on 20 MUTAGENICITY graphs (each with at least one triangle, 22 triangle targets total). O1/O2/O3 cosines are reported as pooled similarities across all supported targets from all 20 graphs.

*Table 13.* Triangle-support sampling ablation (20 Mutagenicity graphs). Adding triangle-aware samples improves O3 cosine from $0.48$ to $0.91$.

| Sampling Config | O1 Cos | O2 Cos | O3 Cos | Train Size |
|---|---|---|---|---|
| O1/O2-aware (base) | 0.978 | 0.889 | 0.477 | 155.7 |
| O1/O2/O3-aware | 0.984 | 0.924 | 0.778 | 232.0 |
| O3-tuned (more tri-flips) | 0.986 | 0.929 | **0.905** | 281.7 |

Adding triangle flip-3 neighbors improves O3 cosine from $0.477$ to $0.905$, and edge-only O2 also improves monotonically ($0.889 \to 0.929$). Uniform sampling at matched budget achieves O3 cosine $0.752$—competitive but below the structure-aware variant. The main change across rows is the training-set composition: the base and edge components stay fixed while triangle-aware masks increase from 0 to 126 on average. These results demonstrate that structure-aware sampling improves the recovery of higher-order interactions, with triangle-aware sampling consistently boosting the cosine similarity of third-order interactions with the ground-truths.

### F.18. Amortization Details

In the low-budget GPU regime (20 matched Mutagenicity graphs, $n \leq 20$), TN-SHAP-G training takes a mean of $4.12\,\text{s}$ (max $6.53\,\text{s}$). Per-query inference costs are for O1 is $0.010$ seconds and for O2 is $0.119$ seconds. GraphSHAP-IQ requires $2.08\,\text{s}$ per O1+O2 query. TN-SHAP-G becomes the faster method at $K \geq 5$ repeated queries.

On the PROTEINS dataset (graphs up to 620 nodes), GraphSHAP-IQ encounters out-of-memory errors on all 4 tested large graphs (return code 137), while TN-SHAP-G succeeds on all 4.

### F.19. Scope of the Locality Assumption

The key assumption is not that all graph models are local, but that the induced coalition game admits *low-rank structure aligned with graph separators*. This is naturally plausible for message-passing GNNs, which are the main focus of our

quantitative experiments. For models with broad attention (e.g., graph transformers), the required bond dimension may increase.

We test the fidelity of the Shapley values on a Graphormer-style model (Appendix F.3). After hyperparameter tuning ($\chi = 12$, 2400 epochs), TN-SHAP-G achieves mean O1 cosine **0.915** across 6 Mutagenicity graphs (up from $0.801$ before tuning), with $25\times$ inference speedup over exact enumeration. The lower test $R^2$ ($0.524$ vs. $> 0.9$ for GINs) reflects the harder game induced by global attention, but Shapley recovery remains meaningful. This suggests the framework extends beyond MP-GNNs when the model has learned local structure from the data.

### F.20. Extending to Edge and Subgraph Players

The framework applies once one specifies (i) the player set $P$ and (ii) the coalition-dependent intervention operator $\mathcal{M}$.

**Edge players.** Set $P = E$. For a coalition $S \subseteq E$, define the intervention by removing edges $E \setminus S$ from the adjacency matrix (or setting their weights to zero): the GNN receives the modified graph $G_S = (V, S)$. No edge features are required; the intervention is purely structural.

**Subgraph/motif players.** Define a set of predefined motifs $\{M_1, \ldots, M_K\}$ (e.g., rings, functional groups). Set $P = \{M_1, \ldots, M_K\}$. For a coalition $S \subseteq P$, mask all nodes not belonging to any motif in $S$. The TN topology mirrors the motif-interaction graph, where two motifs are connected if they share nodes or edges.

## G. Additional related work and discussion (extended)

**Shapley interactions and efficient estimators.** Higher-order interactions have been formalized through the Shapley–Taylor index (Sundararajan & Najmi, 2020), faith-Shapley (Tsai et al., 2023), and efficient estimators such as SHAP-IQ (Fumagalli et al., 2024). Surveys such as Yuan et al. (2022) provide comprehensive overviews of graph explanation methods.

**Sample complexity of tensor network hypothesis classes.** Recent results bound the pseudo-dimension of tensor network hypothesis classes in terms of parameter count, implying sample complexity scaling of the form $\tilde{O}(N_{\mathcal{G}}/\varepsilon^2)$ for $\varepsilon$-accurate learning (Khavari & Rabusseau, 2021). These bounds motivate learning surrogates with small parameter counts via topology alignment.

