# OpenReview forum: "TN-SHAP-G: Graph-Structured Tensor Network Surrogates for Shapley Values and Interactions"
_ICML.cc/2026/Conference — ICML 2026 regular_

### Official Review · Reviewer_R5Qj · 2026-03-02

**Soundness:** 3
**Presentation:** 3
**Significance:** 3
**Originality:** 4
**Overall Recommendation:** 5
**Confidence:** 4

**Summary:**

The paper proposes a novel way of computing **exact** Shapley Values for a new structure of learning model, a tensor network (TN), which can be used as a surrogate for different kinds of black-box models. The paper focuses on graph-level tasks. Therein, the paper shows how a graph-aligned tensor network can be fitted on a masked graph game and then analytically compute the exact SVs/SIs from the tensor network.

**Summary:** A well-executed paper where some more work could have been put into showing the generality of the TNs and where they can be applied effectively. However the contribution is already quite strong warranting acceptance.


### References (used throughout the review):
1. https://openreview.net/forum?id=Qabko39AS5
2. https://openreview.net/forum?id=KI8qan2EA7
3. https://papers.nips.cc/paper_files/paper/2022/hash/a5a3b1ef79520b7cd122d888673a3ebc-Abstract-Conference.html
4. https://www.nature.com/articles/s42256-019-0138-9
5. https://ojs.aaai.org/index.php/AAAI/article/view/26322
6. https://ojs.aaai.org/index.php/AAAI/article/view/29352

**Compliance With Llm Reviewing Policy:**

Affirmed.

**Final Justification:**

While being rather incremental and only focused on graph-based tasks this paper still is very well executed and presented. The method improves on the state-of-the-art in a meaningful way such that people interested in this research field will benefit from this submission. The rebuttal showed me actually that the paper is a bit more limited/incremental than I read it as in the beginning. Nonetheless my score increased since my concerns have been adequately addressed and I do not see the more niche focus of the paper as an obstacle.

**Key Questions For Authors:**

- Q1: Can this new way of computing Shapley Values exactly be applied as a new surrogate/proxy in the Regression MSR principle outlined in [1]?
- Q2: How easy is the fitting of your tensor networks? Can you elaborate on computational costs?
- Q3: How adaptable is the method? Why did you only consider the graph setting? Could this not also be applied to any black-box machine learning problem or are there no substantial performance gains?
- Q4: What exact type of interactions do you compute? I think the paper lacks detail there a bit (Weakness 3). The paper cites FaithSHAP and the Shapley interaction index, but I did not really spot which is used here. I assume the Shapley interaction index.
- Q5: How does the polynomial structure presented here refer to the polynomial representations in Linear TreeSHAP [6] and TreeSHAP-IQ [6]

**Limitations:**

The paper does not explicitly state its main limitations. However, I also do not necessarily miss anything here.

**Strengths And Weaknesses:**

### Strengths
- **Contribution**: The clear strength of this paper is a novel and valuable contribution to the computational theory of Shapley Values and interactions. The paper shows that the exact computation of such summary scores is achievable for a special case of Tensor Networks (TNs). New **exact** computation schemes for the Shapley Value are extremely valuable as these methods are becoming more and more important. This has a lot of direct applications and room for interesting future work, which I also think this is the weakest aspect of the paper which probably can be improved (see Weakness 2 and Question 1).
- Very good presentation.
- paper is well written and clear. Good job.
- Code is well presented and should be no real problem to reproduce (see Question 2).
- While there are things missing (see Weakness 1), the related work already contains a lot of important baselines. I also appreciate the detailed comparison with GrapSHAP-IQ wich is a recent method presented for the same domain used here.

### Weaknesses
1. Quite poor related work on the Shapley side: Recently, a new paradigm for model-agnostic estimation of Shapley Values has been presented: The Regression MSR principle [1] also part of ProxySPEX [2]. This could have been a direct application for the here presented contribution, which would have been very interesting in the context discussed here. Also, the whole line of work surrounding TreeSHAP [3-6] and its extensions are missing, which this work is quite related to. Particularly the work surrounding TreeSHAP and the polynomial representation for linear TreeSHAP is very related to this paper's contribution. A more robust comparison to this line of work would make the paper a bit more aligned with the Shapley literature improving its fit.
2. Weak baselines and comparisons: While it is not paramount for this paper to show better performance than model-agnostic approximations per-se (the exact computation is the focus), it would have been an improvement of this work to contrast it more to the current state-of-the-art. For example, I think it would have been interesting to contrast/position the contribution with the Regression MSR principle of using a stronger model family (e.g. Trees) for modeling the value function and then applying interventional TreeSHAP to get the exact Shapley Values for the proxy model. Then the exact Shapley Values are used to create a residual game for which an unbiased/consistent estimator *adjusts* the estimates. Please also see/respond to Question 1 here.
3. The paper lacks detail in the discussion of what kind of Shapley Interactions are used and computed. The paper cites both the Shapley interaction index and FaithSHAP (and its FaithSHAP interaction extension). For the experiments, it is not really clear what interactions are computed. A bit more detail would be helpful.

---

> ### Author Rebuttal · Authors · 2026-03-31
>
> Thank you for your positive assessment. We clarify below how we will better position our contribution wrt the broader Shapley literature in the revision. We first want to stress that **TN-SHAP-G is not simply fitting a generic regressor to coalition values**. It learns a **tensor-network parameterization of the multilinear extension** of a cooperative game. This matters because the multilinear extension is the object from which **Shapley values and interaction indices** are recovered. While multilinear extensions of games are classical in game theory, they remain underexplored in mainstream ML explainability: TN-SHAP [1] and TN-SHAP-G are the first methods to leverage the low rank structure of the multilinear extension. TN-SHAP-G also makes the multilinear extension representation graph-aligned, so the approximation class reflects the structure of the graph-induced game, leading to a better inductive bias and more efficient and robust Shapley approximations.
>
> **W1 (related work: Regression MSR / ProxySPEX / TreeSHAP).** We thank the reviewer for these pointers. With the exception of **GraphSHAP-IQ**, these methods were not developed for graph-structured inputs, and none of them introduce a **graph-aligned multilinear approximation class** for cooperative games on graphs. They are conceptually close in that they also exploit structure or approximation to make Shapley computation more efficient. We will add this discussion and references in the revised related work.
>
> For Regression MSR, the key distinction is that it fits a proxy to the value function and then uses a residual-correction scheme around that proxy. By contrast, **TN-SHAP-G learns a graph-aligned TN representation of the multilinear extension itself**. We will clarify this in **Section 5** and position MSR/ProxySPEX as important related directions. We also agree these perspectives are complementary: in principle, the learned TN multilinear extension could serve as the proxy in an MSR-style pipeline, and there may be multiple correction strategies built around such a multilinear representation. We view this as promising future work, but it is outside this submission.
>
> For TreeSHAP / TreeSHAP-IQ, the connection is high-level. These methods derive exact computation schemes from the combinatorial structure of tree models, often via dynamic programming or polynomial arithmetic tied to tree paths. TN-SHAP-G instead exploits a **low-rank factorization of the multilinear extension** of the coalition game, and in this paper adapts that factorization specifically to graphs through **graph-aligned TN topologies**. Both viewpoints involve polynomial structure, but there is no direct mathematical equivalence.
>
> **Q1 (TN as proxy in MSR).** This is interesting future work that we will mention in the revision, but outside the scope of the current paper. Also addressed in **W1**.
>
> **Q2 (fitting / computational cost).** TN-SHAP-G pays a one-time **instance-specific fitting cost**, after which all desired Shapley quantities can be extracted from the same learned **TN multilinear extension**. On **20 matched Mutagenicity graphs** (n≤20), mean GPU timing is **4.12s** for training, **0.010s** for O1 extraction, and **0.119s** for O2 extraction. We will make this amortization benefit more explicit in the main text. See **Fig 7**, for more details on scalability of training time.
>
> **Q3 (why only graphs?).** The method is not limited to graphs in principle. More generally, it applies whenever one can define a scalar cooperative game and a low-rank multilinear representation is plausible. In that sense, **TN-SHAP [1]** is the more general black-box framework already cited in the paper, while **TN-SHAP-G** is the graph-specific variant introduced here. We focus on graphs because this is where the design is especially well motivated: the input graph itself provides a natural inductive bias for the TN topology. We will clarify this relationship more explicitly.
>
> **Q4 (interaction index).** We compute the **Shapley interaction index**. We will state this directly in **Section 2.2** and the experiments.
>
> **Q5 (relation to polynomial structure in TreeSHAP / TreeSHAP-IQ).** In our case, the multilinear extension is a **multi-affine polynomial**, and the TN provides a **low-rank factorization** of that polynomial. Also addressed in **W1**.
>
> **W2 (baselines / generality).** We will add new experiments to strengthen generality: baseline sensitivity across three baselines (all **O1 >= 0.976**), node-removal semantics (**O1 cosine 0.9680**), and a continuous-feature synthetic benchmark (**O1 cosine 0.994**). These further support that TN-SHAP-G approximates the multilinear extension of a chosen cooperative game, rather than relying on one narrow masking convention.
>
> We hope these clarifications and additions address the reviewer’s concerns.
>
> [1] Heidari, F., Li, C., & Rabusseau, G. (2025). Tractable Shapley Values and Interactions via Tensor Networks.

---

> > ### Author Rebuttal · Reviewer_R5Qj · 2026-04-01
> >
> > I thank the authors for their clarifications. My concerns have been addressed. I have considered adjusting my score. After reading all the remaining reviews, I still think this paper to be between borderline and accept. It is a good contribution but for a limited audience.
> >
> > Additional side comment for the authors: If you compute SII you can also compute the k-SII scores [1] directly. For this there are functions in the shapiq library [2]. k-SII is from an interpretation point of view much more "interpretable" in my opinion and closely resembles the other Faithful Interaction index [3].
> >
> > [1]: https://proceedings.mlr.press/v206/bordt23a.html
> > [2]: https://proceedings.neurips.cc/paper_files/paper/2024/file/eb3a9313405e2d4175a5a3cfcd49999b-Paper-Datasets_and_Benchmarks_Track.pdf
> > [3]: https://jmlr.org/papers/v24/22-0202.html

---

### Official Review · Reviewer_8zap · 2026-03-12

**Soundness:** 3
**Presentation:** 3
**Significance:** 2
**Originality:** 2
**Overall Recommendation:** 3
**Confidence:** 3

**Summary:**

This paper aims to tackle the problem of computing Shapley values on a graph-based model using tensor networks. Unlike traditional approaches, this work introduces a graph-aligned tensor network representation, whose structure is aligned with the input graph data, to represent the Shapley values of the model. Theoretical justifications for this choice are provided, and experiments on several datasets demonstrate the effectiveness of the proposed method.

**Compliance With Llm Reviewing Policy:**

Affirmed.

**Final Justification:**

While this paper is technically solid and well presented, the idea is still rather incremental.

**Key Questions For Authors:**

See Weaknesses.

**Limitations:**

yes.

**Strengths And Weaknesses:**

Strengths

1. Improving the interpretability of machine learning models is important. This paper makes an important contribution to addressing this problem, especially regarding the efficient representation of Shapley values using tensor networks in graph-based models, an area that remains relatively underexplored.

2. Overall, the paper is well structured and technically sound, with theoretical guarantees on model choice and sample complexity, supported by experimental validation.

Weaknesses

1.  How to represent Shapley values using tensor networks has been studied [1]. The main distinction of this paper is that it considers graph-based models, which therefore require a different type of tensor network aligned with the input data structure to improve performance, the underlying idea is not new.

2. Several proofs in the work, particularly in Section 3.1, aim to verify the importance of selecting a tensor network model aligned with the input data structure. As I understand, the core message is that an inappropriate model choice will lead to an explosion in bond dimension. Conversely, a properly aligned model yields better expressiveness. This perspective is known in tensor networks and does not seem to offer many new insights.

Reference


[1] Heidari, F., Li, C., & Rabusseau, G. (2025). Tractable Shapley Values and Interactions via Tensor Networks. arXiv preprint arXiv:2510.22138.

---

> ### Author Rebuttal · Authors · 2026-03-31
>
> We thank the reviewer for their comments. The relationship to **TN-SHAP** will be stated more explicitly in the revision. Our claim is not that using tensor networks for Shapley computation is new in general. Rather, the contribution here is to introduce and analyze a **graph-aligned TN multilinear extension** for graph-induced coalition games, and to show that leveraging structure of TNs aligned with the input data is both theoretically justified and practically useful for estimating Shapley on graphs.
>
> **W1 (novelty / “natural extension to graphs”).** We agree TN-SHAP-G builds on the TN-SHAP [1] line of work. However, we respectfully disagree that the present contribution is merely a routine extension. **Graph alignment is an important design choice.** One could apply TN-SHAP with a generic chain/TT topology to graph-induced games. Our contribution is to show that matching the TN topology to the input graph is beneficial both theoretically and empirically. **Theorems 3.2–3.4** show that graph-aligned TNs admit stronger expressivity/parameter tradeoffs for graph coalition games than generic chain factorizations, and **Fig. 6** confirms this empirically: graph-aligned TNs outperform TT under matched parameter budgets, while TT is additionally sensitive to node ordering. Thus, the main point is not simply “extend TN-SHAP to graphs,” but to show that the structural flexibility of tensor networks can be exploited to build more efficient multilinear representations of graph-induced games, improving both computational and sample efficiency.
>
> We also note that such domain-specific extensions are common in graph explainability: **GraphSVX [2]**  extends **KernelSHAP [3]**  to graphs, and **GraphSHAP-IQ [4]**  extends **SHAP-IQ [5]** to graphs. These extensions are often nontrivial because one must decide how graph structure enters the cooperative game, the perturbation semantics, and the approximation class. In our setting, this is especially important because TNs provide a flexible family of factorization topologies, and the approximation class itself can be matched to the structure of the graph-induced game. At the same time, it is not automatic that this choice should help in practice; one must still show that it improves the approximation of the multilinear extension from sampled coalitions. This is precisely what our theory and experiments aim to establish.
>
> **W2 (Section 3.1 / known TN perspective).** We appreciate this point. While the underlying intuition may be natural from a tensor-network perspective, we believe these results remain important here for two reasons. First, the theorems are instantiated for the coalition tensor of graph-induced games, including the cut-rank/separator view and the graph-vs-TT implications in **Theorems 3.2–3.4**. Second, the graph explainability audience is not typically assumed to be familiar with TN expressivity arguments, and one goal of the paper is precisely to make these tools accessible in the graph Shapley setting. In that sense, these results are meant to justify why graph alignment is not just an implementation detail but a principled modeling choice. We will revise the Introduction and **Section 3.1** to distinguish more sharply between what is inherited from TN theory and what is specific to graph-induced coalition games in our setting.
>
> In the revision, we will state the relationship to **TN-SHAP [1]** explicitly and clarify that the contribution is:
> (a) introducing **graph-aligned TN multilinear extensions** for graph-induced coalition games,
> (b) showing their expressivity advantages over generic topologies, and
> (c) demonstrating their practical effectiveness on graph benchmarks where existing methods fail to scale.
>
> Lastly, we believe our theoretical results are important, in addition to our experiments, to demonstrate to the explainability & graph community that TN are very efficient and versatile tools with a lot of potential for explainability of graph models.
> We hope this clarification addresses the reviewer’s originality concern.
>
> [2] **GraphSVX**
> Duval, A., & Malliaros, F. D. (2021). GraphSVX: Shapley Value Explanations for Graph Neural Networks. In Machine Learning and Knowledge Discovery in Databases (ECML PKDD 2021, Part II), 302–318. Springer.
>
> [3] **KernelSHAP**
> Lundberg, S. M., & Lee, S.-I. (2017). A Unified Approach to Interpreting Model Predictions. In Advances in Neural Information Processing Systems 30 (NeurIPS 2017).
>
> [4] **GraphSHAP-IQ**
> Fumagalli, F., Muschalik, M., Frazzetto, P., Strotherm, J., Hermes, L., Sperduti, A., Hüllermeier, E., & Hammer, B. (2025). Exact Computation of Any-Order Shapley Interactions for Graph Neural Networks. arXiv:2501.16944.
>
> [5] **SHAP-IQ / shapiq**
> Muschalik, M., Fumagalli, F., Baniecki, H., Kolpaczki, P., Hüllermeier, E., & Hammer, B. (2024). Shapiq: shapley interactions for machine learning. In Advances in Neural Information Processing Systems 37 (NeurIPS 2024).

---

> > ### Author Rebuttal · Reviewer_8zap · 2026-04-04
> >
> > Thanks for the rebuttal. The response clarifies the contribution of the work, but the level of novelty remains somewhat limited.

---

### Official Review · Reviewer_7UHo · 2026-03-12

**Soundness:** 3
**Presentation:** 3
**Significance:** 4
**Originality:** 3
**Overall Recommendation:** 5
**Confidence:** 3

**Summary:**

Motivated by explainability issues stemming from graph based data, the paper derives a deterministic node attribution algorithm based on Shapley computation by learning a graph aligned tensor network surrogate from a small number of oracle queries. The computational complexity is linear in graph size and they also show that exact deterministic attribution is possible with O(n) queries. The linear complexity is made possible by exploiting the low rank structure in its value functions. Their claim is supported by rigorous theoretical guarantees. They also verify their claims using molecular benchmarks beating other known methods.

**Compliance With Llm Reviewing Policy:**

Affirmed.

**Final Justification:**

I remain positive on this paper and keep my original score. My main reason for supporting acceptance is that the paper makes a strong and practically meaningful contribution to graph explainability. In an area where many existing approaches rely on expensive Monte Carlo estimation or heuristic approximations, this is a substantial step forward. The empirical results on molecular benchmarks also support the practical value of the method.

I view this as a technically solid, original, and significant paper whose limitations are understood and whose main contribution remains strong. For these reasons, I continue to recommend acceptance.

**Key Questions For Authors:**

The proposed method relies on exploiting locality structure in the coalition value function. Could the authors clarify a bit more under what conditions this locality assumption holds and when it might fail? For instance, while message-passing GNNs may naturally exhibit local interactions as mentioned in the paper, for which classes of models might this assumption break down? Providing concrete examples would help clarify the scope and limitations of the proposed approach.
Is the sampling from the full coalition space straightforward? Are there points that a casual reader should be more aware of?

**Limitations:**

yes

**Strengths And Weaknesses:**

Strengths:
Computing exact Shapley values for a general cooperative game is notoriously difficult and in practice most approaches rely on Monte Carlo sampling or other heuristic approximations. In the context of graph explainability many existing methods require hundreds or thousands of model evaluations to estimate node attributions. The paper addresses this limitation directly by proposing a deterministic attribution algorithm with linear complexity in the graph size. This significantly improves the scalability of Shapley-based explanations and represents a meaningful contribution to the community.
The approach exploits the low-rank structure of the value function by learning a graph-aligned tensor network surrogate from a small number of oracle queries. This structural insight enables efficient deterministic attribution while maintaining theoretical soundness.
The authors support their claims with rigorous theoretical guarantees while keeping the presentation reasonably accessible. In addition, the empirical evaluation on molecular graph benchmarks demonstrates the practical effectiveness of the approach and shows improvements over existing methods.

Weaknesses:
Some of the theoretical results are presented only as proof sketches. Since the appendix does not appear to have strict page limits, it would improve the paper’s clarity and completeness if the authors included full proofs for these results. This would allow readers to more easily verify the claims and better understand the underlying arguments.

---

> ### Author Rebuttal · Authors · 2026-03-31
>
> Thank you for your evaluation and constructive comments.
>
> **W1 (proof completeness)**
>
> Thank you for this suggestion. Most of the main technical results have complete proofs in the appendix except for **Corollaries 3.3, 3.5 and Thm 3.6**.
> Corollary 3.3 is a direct consequence of **Thm 3.2**, we will explicitly write in the main paper.
> The proof of **Corollary 3.5** is indeed only given as a sketch in lines 263-274, we will add a complete proof in the revised appendix.
> We will also add a complete proof of how **Thm 3.6** is derived from **Thm 4 in (Khavari and Rabusseau)** in the revised appendix.
>
> **Q1 (locality assumption).** The intended assumption is not that all graph models are local. Rather, it is that the induced coalition game admits some **low-rank structure aligned with graph separators**. This is naturally plausible for **message-passing GNNs**, which are the main focus of our quantitative experiments. It may weaken when the model or task induces stronger global interactions, e.g. in graph-transformer-style settings with broad attention; in such cases, one may need larger bond dimension, richer TN topologies, or graph rewiring so that the TN better matches the true interaction structure. We will clarify this scope more explicitly in **Sec. 3.1**.
> Even though the locality assumption is not structurally enforced in graph transformers, learned models can capture this structure from the data, This is what our qualitative Graphormer-style experiment (**Appendix F.3 / Fig. 11**) shows: TN-SHAP-G performs well even to approximate Shapley of graph transformers when they learned the local structure in the data. Of course, TN-SHAP-G could fail in cases where the model has learned very long range dependencies.
> We will point to this result alongside the locality discussion in the revision to show that the method also applies beyond message-passing GNNs, while keeping the main quantitative focus of the paper on the message-passing setting.
>
> **Q2 (coalition sampling).** We agree this can be explained more clearly. The paper already contains a sampling-strategy ablation in **Fig. 13 / Appendix F.5**, including uniform and structure-aware variants; the main takeaway is that uniform sampling already works reasonably well, while structure-aware sampling improves **O2/O3 recovery**. We also ran a new triangle-support ablation on **20 Mutagenicity graphs**, which confirms the same message: adding triangle-aware samples improves higher-order recovery at modest additional cost. We will summarize this intuition more clearly in **Sec. 3.2** and keep the implementation details in the appendix.
>
> We hope these answers addresses your concerns and thank you again for your questions which will help us clarify important points in the revision.

---

> > ### Author Rebuttal · Reviewer_7UHo · 2026-04-01
> >
> > Thank you for the rebuttal. The response addresses my questions well. In particular, the clarification that the key assumption is low-rank structure aligned with graph separators, is helpful, as are the concrete examples of settings where the approach is expected to work well versus where it may weaken. The discussion of coalition sampling is also useful, especially the clarification that uniform sampling already works reasonably well while structure-aware sampling can improve higher-order recovery. I also appreciate the commitment to add the missing full proofs in the appendix. I will keep my original score.

---

### Official Review · Reviewer_by3f · 2026-03-16

**Soundness:** 3
**Presentation:** 1
**Significance:** 2
**Originality:** 4
**Overall Recommendation:** 5
**Confidence:** 3

**Summary:**

The paper proposes TN-SHAP-G to calculate node Shapley values for graph neural networks. The authors define the cooperative game by masking node features, wherein a node is masked by replacing its features with the average of the node features of all other nodes in the graph. Direct Shapley value calculation on a graph of $n$ nodes is prohibitively expensive as it requires summing over $2^{n-1}$ coalitions. The authors show that the multilinear extension of Shapley values can be written as a tensor contraction between a coalition-value tensor and Bernoulli selector vectors, enabling approximation of the game using tensor networks. This reduces computation while enabling exact Shapley value calculation. However, the coalition tensor has $2^n$ entries, requiring $2^n$ GNN calls. To combat this, the authors train a surrogate tensor network as a regression problem over the value function $v(S)$, where $v(S)$ is defined as the logit of the predicted class under coalition $S$. The surrogate is trained to match $v(S)$ for those $M$ coalitions, where $M \ll 2^n$, sampled from a mixture distribution to cover a wide range of coalitions. The surrogate implicitly represents the $2^n$ coalition values while storing them with far fewer parameters by factorizing the interaction structure. Once this surrogate tensor network is trained, it implicitly represents the multilinear extension of the coalition game, enabling the computation of Shapley values and interaction indices deterministically via derivatives and polynomial interpolation, without additional model queries.

**Compliance With Llm Reviewing Policy:**

Affirmed.

**Final Justification:**

The authors have addressed my concerns satisfactorily. I have increased my score from "Weak reject" to "Accept".

**Key Questions For Authors:**

Please elaborate this:

> Section 2, line 72: "Although we focus on node-level explanations (players $v \in V$), the same construction applies when players correspond to edges or subgraphs by redefining the masking operator accordingly."

Since TN-SHAP-G masks nodes by replacing their features with a baseline value, does masking edges require edges to have features? Or does it take two nodes at a time, similar to App D.3.1? What happens at the subgraph level?

**Limitations:**

See weakness 9 on categorical datasets.

**Strengths And Weaknesses:**

# Strengths
1. The conversion of the combinatorial coalition game into a tensor-network representation is novel.
2. The framework supports higher-order Shapley interactions.
3. The paper comes with a lot of theoretical rigor.
4. The authors have shared their code along with an instructional README.

# Weaknesses
**1. Questionable value function formulation**
- The paper defines the exclusion mechanism in the Shapley game through **node feature masking rather than node removal or structural perturbation**. Specifically, nodes outside the coalition remain in the graph but have their features replaced by a baseline vector. As a result, masked nodes still participate in message passing and influence the prediction through graph topology.
- For models that rely on structural signals (e.g., degree information, connectivity patterns, or positional encodings), this masking scheme does not truly remove the contribution of a node. Instead, it primarily removes the node's feature information while preserving its structural role in the graph.
- Since the central objective of the pipeline is to accurately approximate the coalition value function, the interpretation of this function becomes critical. **Under the proposed masking scheme, the value function effectively answers the question: "Which nodes' features matter?" rather than "Which nodes matter?"**, which is arguably the more natural interpretation of node-level Shapley values in graph settings.
- Consequently, the coalition game being approximated does not reflect the intended semantic notion of node importance for GNNs.

**2. Unrealistic baseline choice**
- The paper replaces masked node features with the mean feature vector of the remaining nodes. While common in tabular SHAP-style methods, this choice may be poorly grounded for many graph domains. For example, in molecular graphs, averaging atom feature vectors does not correspond to any physically meaningful atom type. This can also produce out-of-distribution inputs.

**3. Choice of scalar value function / Surrogate model's training**
- The coalition value is defined as the logit of the predicted class produced by the black-box GNN. The surrogate model is trained to regress this scalar quantity. This ignores interactions between class logits and therefore does not capture the full predictive landscape of the model. As a result, the shapley values reflect changes in a single logit rather than the model's overall decision dynamics across classes.

**4. Per-instance surrogate training**
- The approach requires training a separate tensor-network surrogate for each graph instance being explained.


**5. Scalability**
- The paper claims that TN-SHAP-G scales efficiently; however, the experimental evidence supporting this claim is limited. The largest graphs evaluated contain at most around 600 nodes, which is relatively small compared to the scale of most real-world graphs. Hence, demonstrating scalability on graphs of this size does not substantiate the claim of practical scalability.

- Additionally, for graphs with more than 100 nodes, the method relies on graph coarsening as a preprocessing step. This introduces an additional approximation layer and further downplays the scalability claims for the original graph.

**6. Slightly misleading terminology**
- The proposed method computes exact Shapley values with respect to the learned surrogate model, rather than with respect to the original black-box GNN. While this distinction is discussed in the paper, some statements give the impression that exact Shapley values of the original model are being computed. A better wording would be preferred:
    - Line 30: _Once trained from a small number of model queries, the surrogate enables exact, deterministic recovery of Shapley values._
    - Github repo readme: _"Deterministic: No Monte Carlo sampling needed for Shapley computation"_

**7. Novelty concerns / Missing baseline**
- The authors missed a close baseline: ICLR 25: Muschalik, Maximilian, et al. _"Exact computation of any-order Shapley interactions for graph neural networks."_
- This work addresses the exact same problem, as evident from its abstract:
    - > "we explain single graph predictions of GNNs with Shapley Interactions that quantify node contributions and interactions among multiple nodes."
    - > "we introduce GraphSHAP-IQ, an efficient approach to compute any-order SIs exactly."
- While the authors do compare against SHAP-IQ (NeurIPS 24), a prior paper from the same group as GraphSHAP-IQ's, that method isn't tailored for graphs, whereas the above work specifically is.
- This authors must articulate their method's novelty and demonstrate experimental superiority over this baseline.

**8. Clarity and Readability**
- The abstract and introduction are overly dense, and the paper's overall readability is poor. Technical jargon is introduced without sufficient context, making it difficult to grasp the core problem early on. In my view, many key concepts of the pipeline cannot be assumed to be familiar to the broader graph research audience. Readers unfamiliar with tensor-related literature would struggle to read this paper.

**9. Datasets with categorical features**
- All the datasets used have categorical features, like atom type. Table 5 does mention a good number of features for each dataset. However, all of them are essentially one-hot-encoded categorical features.
- The paper's prelims mentions that $X \in \mathbb{R^{n \times d}}$. Hence, it should admit graphs with rich features.
- If this is a limitation, then it should be mentioned explicitly for the reader's convenience.

**Minor comments:**
- Table 5 mentions the MUTAG dataset, but the paper doesn't have results on MUTAG, and its description is also missing from section F8. It is best to drop it from the table.
- The text suggests the presence of additional datasets beyond those actually used. For example, Section 4 (Line 366) states: _"...evaluate on standard molecular benchmarks including Mutagenicity, Benzene, and proteins."_ However, these are the only datasets used.

---

> ### Author Rebuttal · Authors · 2026-03-31
>
> We thank the reviewer for recognizing the **novelty** of our TN formulation to handle **higher-order interactions**, and its **theoretical rigor**. A key clarification, relevant to **W1–W3 and W9–Q1**, is that **TN-SHAP-G is not a generic regressor on raw graph features**. The graph, node features, and masking/intervention semantics enter only through the black-box predictor to define a scalar coalition game. TN-SHAP-G then learns a graph-aligned tensor-network parameterization of the **multilinear extension** of this game as a function of binary coalition indicators. Thus, the TN learns how the model output varies over subsets of players; it does not take raw features directly as input. This is why the framework is agnostic to feature type and intervention semantics. We agree the manuscript did not make this separation between the game definition and the TN approximation of its multilinear extension explicit, and we will foreground it in the revision.
>
> **W1–W2 (game semantics / baseline choice).** We add experiments showing robustness across different baselines, node removal masking, and different game semantics: Across settings, TN-SHAP-G remains robust: **O1 cosine is 0.997/0.976/0.998 under mean/zero/ones baselines, 0.968 under node-removal semantics, and 0.965 (O1) / 0.969 (O2) under Myerson game semantics.**
>
> These results show that TN-SHAP-G is not tied to a single baseline, masking rule, or attribution semantics.  We will revise **Sec. 2.1** to make explicit that game semantics are a modeling choice orthogonal to the TN framework and add the experiment above to the experiment section .
>
> **W3 (scalar value function).** Following standard practice as in GraphSHAP-IQ, we use **predicted-class logit** for the value function. We note that TN-SHAP-G is not tied to logits and can be applied to any other **(scalar) value function** derived from the logits.
>
> **W4 (per-instance training).** Correct: TN-SHAP-G learns an **instance-specific TN multilinear extension**. The relevant comparison is not one training run versus one attribution query; it is one training run versus the cost of repeatedly estimating many Shapley quantities on the same instance. Once trained, the same TN amortizes all quantities for that instance: **O1 values, higher-order interactions, and node-specific synergies** can all be extracted from the same learned TN. On 20 matched Mutagenicity graphs, mean TN training is **4.12s**, **O1 extraction 0.010s**, and **O2 extraction 0.119s**. Relative to exact GraphSHAP-IQ, this already yields amortized total-time wins on average at **K=5** queries. We will add an amortization comparison table with GraphSHAP-IQ in the main text and refer more explicitly to  **Fig. 7**.
>
> **W5 (scalability).** Even though **600 nodes** may not seem large in the context of graph learning, most Shapley methods become infeasible well below this.  GraphSHAP-IQ encounters out of memory errors above **~40-60 nodes** (**App. F.12, Table 7**), and sampling based methods exhibit high variance even with tens of thousands of queries. TN-SHAP-G can compute very accurate approximations for Shapley values and interactions for graphs twice as large with **$R^2 > 0.98$**. Coarsening (**App. F.7**) pushes this further to **600+ nodes** with **$R^2 > 0.8$** and **Shapley efficiency gaps under $10^{-5}$** (**Fig. 10**). For the largest graph we ran (**n=620**), permutation sampling and SHAP-IQ each require **~21,700 queries per seed**, with multiple seeds needed to reduce variance. TN-SHAP-G requires only **10,000 queries from a single run**, with no repeated seeds, while amortizing all first- and second-order indices from the same TN.
>
> **W6 (“exact” wording).** We will revise the wording throughout. The correct statement is: **exact with respect to the learned TN multilinear extension**.
>
> **W7 (GraphSHAP-IQ).**
> This comparison is already included in **Sec. 4 / App. F.12 (Table 7)**. Also more details provided in **W5**. We will also add a runtime/accuracy comparison in the small-graphs as discussed in **W4**.
>
> **W8 (readability).** We will simplify the **Abstract/Introduction**, define the main objects earlier, and add a short **TN primer** in the appendix.
>
> **W9 (categorical features).** This is not a restriction of the method. We will clarify that **PROTEINS**, already used in our scaling experiments, contains **continuous node attributes**, and we also add a **synthetic continuous-feature experiment**.
>
> **Q1 (edge / subgraph players).** The framework applies once one specifies **(i) the player set and (ii) the coalition-dependent intervention operator**. Thus, edge explanations can use structural interventions without requiring edge features, and subgraph explanations can use predefined motifs/subgraphs as players. We will add a short appendix recipe clarifying this.
>
> **Minor comments.** We will also make the dataset wording fully consistent.
>
> We hope these clarifications and new experiments address the reviewer’s concerns.

---

> > ### Author Rebuttal · Reviewer_by3f · 2026-04-03
> >
> > The authors have addressed my concerns satisfactorily. I have increased my score from "Weak reject" to "Accept".

---

### Decision · Program_Chairs · 2026-04-30

**Decision:**

Accept (regular)

**Comment:**

TN-SHAP-G provides a technically rigorous approach to graph explainability by learning graph-aligned tensor network surrogates to compute Shapley values with linear complexity. This method effectively amortizes the cost of high-order interaction queries and demonstrates superior scalability on molecular benchmarks compared to traditional sampling-based methods.

Through the rebuttal, the authors proved that graph-aligned topologies offer unique expressivity advantages over generic factorizations. While one reviewer remained concerned about the incremental novelty relative to general tensor network theory, the majority was convinced by the method's theoretical grounding and empirical robustness. The significant score upgrade from one reviewer further underscores the effectiveness of the authors' clarifications.